# ROBUST LOCALLY DIFFERENTIALLY PRIVATE GRAPH ANALYSIS

## ABSTRACT

Locally differentially private (LDP) graph analysis allows private analysis on a graph distributed across multiple users. However, such computations are vulnerable to poisoning attacks, where an adversary can skew results by submitting malformed data. This paper studies the impact of poisoning attacks on graph degree estimation protocols under LDP. We make two key contributions. First, we observe that LDP makes protocols more vulnerable to poisoning – the impact is worse when adversaries poison their (noisy) responses rather than their input. Second, we note that graph data is naturally redundant, as every edge is shared between two users. Leveraging this redundancy, we design robust degree estimation protocols under LDP that reduce the impact of poisoning and compute accurate degree estimates. We evaluate our protocols on real-world datasets to demonstrate their effectiveness.

## 1 INTRODUCTION

A distributed graph is defined over a set of users, where each user only knows the edges involving them—in other words, each user has access to their own adjacency list. This means each user has a local view of the graph, and no single entity has knowledge of the entire graph. A real-world example of this can be found in decentralized social media platforms, such as Mastodon, Diaspora, PeerTube, where each user (account holder) represents a node, and an edge between two users indicates they are "friends" (i.e., they follow each other). In this scenario, an untrusted aggregator, such as Mastodon itself, may attempt to compute statistics for the entire graph. However, since the edges represent sensitive information (e.g., edges reveal users' personal social connections), users cannot submit their data to the aggregator directly. Instead, they add noise to their data to achieve a local differential privacy (LDP) guarantee before sharing it with Mastodon. LDP has already been deployed by major commercial organizations such as Google Erlingsson et al. (2014), Apple Greenberg (2016), and Microsoft Ding et al. (2017).

The distributed nature of LDP, however, makes it vulnerable to poisoning attacks. For instance, it is both easy and realistic for an adversary to inject fake users into the system (e.g., by creating fake accounts on Mastodon) or compromise the accounts of real users (by hacking) to run untrusted applications on user devices. Consequently, there is no guarantee that these users will comply with the LDP protocols. The adversary can send carefully crafted malformed data from these non-compliant users and skew estimates, including those involving only honest users.

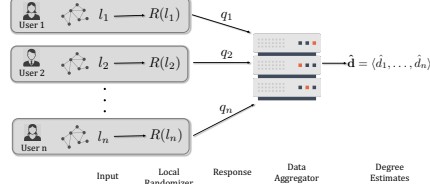

Figure 1: Analysis in the LDP setting

Prior work, which focuses on tabular data Cheu et al. (2021); Cao et al. (2021); Li et al. (2022), finds that poisoning attacks can be carried out against LDP protocols. However, the impact of poisoning under LDP for graph analysis is largely unexplored. In this paper, we initiate a formal study on the impact of poisoning on LDP protocols for graph statistics. We focus on the task of degree vector estimation, one of the most fundamental tasks in graph analysis Raskhodnikova & Smith (2016).

A real-world use-case for a poisoning attack is as follows – suppose a company is interested in hiring the most influential nodes (users) of a graph for marketing its product on Mastodon and uses a node's degree as its measure of influence. An adversary might want to promote a specific malicious node to

be selected as an influencer or prevent an honest node from being selected as an influencer; concretely, suppose a single malicious user wants to be selected as influential. If the LDP protocol used is the Laplace mechanism, where each user directly submits their (noisy) degree to the analyst, then the malicious user can lie flagrantly and report their degree to be $n - 1$, the maximum degree possible!

We address this challenge and design degree estimation protocols that are robust to poisoning attacks. Our algorithms are based on the key observation that graph data is naturally redundant – the information about an edge $e_{ij}$ is shared by both users $U_i$ and $U_j$. Importantly, the users do *not* explicitly share this information; rather, it is implicitly shared by the structure of the graph itself. For example, in a social media graph, both users are aware of their mutual "friend" connection (i.e., the edge between them). Leveraging this observation, we propose robust protocols based on two new ideas. First, we use *distributed information* – we collect the information about each edge from *both* users. The second idea is to *verify* that the collected information is consistent. Specifically, as long as at least one of $U_i$ or $U_j$ is honest, the analyst can check for consistency between the two edge reports and detect malicious behavior.

A key challenge to our mechanism design is that LDP forces all consistency checks to be probabilistic— edge inconsistencies may arise from both malicious behavior and random noise. Consequently, our protocols will flag malicious users using confidence intervals based on the expected number of inconsistent edges. We must set these intervals precisely, so that malicious users who fabricate many edges are caught, while honest users whose users are never falsely flagged as malicious. This requires us to carefully define what it means for a protcol to behave robustly. In summary, we are the *first* to study the impact of poisoning on LDP degree estimation for graphs. Our main contributions are:

- **Novel Formal Framework.** We propose a formal framework for analyzing the robustness of a protocol. Specifically, we measure the robustness along two dimensions, *accuracy* (for honest users) and *soundness* (for malicious users). Intuitively, good accuracy means that the protocol has high utility for honest users, and good soundness means that it can detect/restrict malicious users.

- **New Results on Poisoning Attacks.** Based on the proposed framework, we study the impact of poisoning on private degree estimation in the LDP setting. The attacks can be classified into two types: (1) input poisoning where the adversary does not have access to implementation of the LDP protocol and can only falsify their input (Fig. 2a), and (2) response poisoning where the adversary can tamper with the LDP implementation and directly manipulate the (noisy) responses of the LDP protocol (Fig. 2b). The former is independent of LDP while the latter utilizes the characteristics of LDP. We observe that LDP makes a degree estimation protocol more vulnerable to poisoning – the impact of response poisoning is worse than that of input poisoning. Additionally, we provide a *lower bound* for input poisoning.

- **Novel Robust Degree Estimation Protocols.** Leveraging the natural redundancy in graph data, we design robust degree estimation protocols under LDP that significantly reduce the impact of adversarial poisoning and compute degree estimates with high utility. Our robustness guarantees are *attack-agnostic* – they work for all attacks on all graphs. Additionally, our results matches with the lower bound (upto constants) for input poisoning.

- **Comprehensive Attack Evaluation.** We conduct a comprehensive empirical evaluation to validate our theoretical results. First, we assess the threat of poisoning attacks through **16** real-world motivated scenarios. Our findings reveal that even a small number of malicious users ($m = 1\%$) can inflict significant damage. Next, we demonstrate the robustness of our degree estimation protocols against these attacks. Our results show that our protocols effectively mitigate poisoning attacks even with a larger number of malicious users ($m = 33\%$) in real-world datasets.

## 2  RELATED WORK

A recent line of work Cheu et al. (2021); Cao et al. (2021); Wu et al. (2021); Li et al. (2022) has explored the impact of poisoning in LDP. However, these works focused either on tabular data or key-value data. Additionally, prior work mostly focuses on the task of frequency estimation which is different from our problem of degree estimation. For the former, each user has some item from an input domain and the data aggregator wants to compute the histogram over all the users' items. Whereas, we compute the degree vector $\langle \hat{d}_1, \ldots, \hat{d}_n \rangle$ which is *not* an aggregate query – each user directly reports their degree $d_i$ (a count or via an adjacency list). A detailed discussion is in App. F. A long line of work has proposed LDP protocols for computing different statistics over a distributed graph Blocki et al. (2012); Chen et al. (2020); Hay et al. (2009); Day et al. (2016); Imola et al. (2022;

2021). However, none of them address the problem of data poisoning. To the best of our knowledge, ours is the first work to study the impact of poisoning for graphs under LDP.

A slew of poisoning attacks Biggio et al. (2012); Mei & Zhu (2015); Fang et al. (2020); Bhagoji et al. (2019); Chen et al. (2017); Bagdasaryan et al. (2018); Xie et al. (2020) on machine learning (ML) models have been proposed in the federated learning setting Kairouz et al. (2019). Note that ML model training is *fundamentally different* from the task of graph analysis. Hence, none of the techniques from this literature are directly applicable here.

## 3 PRELIMINARIES

We focus on the privacy guarantee of *edge* LDP Nissim et al. (2007); Raskhodnikova & Smith (2016) which protects the existence of an edge between any two users. In other words, on observing the output, an adversary cannot distinguish between two graphs that differ in a single edge. Edge LDP is the most popular and standard notion of privacy for distributed graphs and has been widely studied widely Imola et al. (2022; 2021); Wang et al. (2016); Qin et al. (2017a); Cormode et al. (2018); Zhang et al. (2020); Xia et al. (2021); Zheng et al. (2021). Formally:

**Definition 1** ($\epsilon$-Edge LDP Qin et al. (2017b)). *Let $R : \{0,1\}^n \mapsto \mathcal{X}$ be a randomized algorithm that takes an adjacency list $l \in \{0,1\}^n$ as input. We say $R$ provides $\epsilon$-edge LDP if for any two neighboring lists $l, l' \in \{0,1\}^n$ that differ in one bit (i.e., one edge) and any output $s \in \mathcal{X}$,*

$$\Pr[R(l) = s] \leq e^\epsilon \Pr[R(l') = s]$$

Randomized Response ($RR_\rho$) Warner (1965) releases a bit $b \in \{0,1\}$ by flipping it with probability $\rho = \frac{1}{1+e^\epsilon}$. We extend the mechanism to inputs in $\{0,1\}^n$ by flipping each bit independently with probability $\rho$ which satisfies $\epsilon$-edge DP. The Laplace mechanism($R_{Lap}$) is a standard algorithm to achieve DP Dwork & Roth (2014). For degree estimation, each user $U_i$ simply reports $\tilde{d}_i = d_i + \eta, \eta \sim Lap(\frac{1}{\epsilon})$ where $Lap(b)$ represents the Laplace distribution with scale parameter $b$. This mechanism satisfies $\epsilon$-edge DP.

### 3.1 PROTOCOL SETUP

**Problem Statement.** We consider single round, non-interactive protocols in which each user $U_i, i \in [n]$ runs the local randomizer $R_i : \{0,1\}^n \to \mathcal{X}$ on their adjacency lists $l_i$. The aggregator collects the noisy responses and applies a function $D : \mathcal{X}^n \to (\mathbb{N} \cup \{\bot\})^d$ to produce final degree estimates $\hat{\mathbf{d}} = \langle \hat{d}_1, \ldots, \hat{d}_n \rangle$. Here, $\hat{d}_i$ is the aggregator's estimate for $d_i$ for user $U_i$. Note that the aggregator is allowed to output a special symbol $\bot$ for a user $U_i$ if they believe the estimate $\hat{d}_i$ to be invalid (i.e., $U_i$ is malicious). Degree vector estimation is one of the most fundamental tasks in graph statistics and is very commonly used for measuring the influence of a node (also known as degree centrality Borgatti & Everett (2006); Kempe et al. (2005)). As ours is the first study to examine the impact of data poisoning on graphs under LDP, we begin by addressing this fundamental problem.

**Threat Model.** In executing protocol $\mathcal{P}$, a subset of users $\mathcal{M} \in [n]$ may be malicious, with $m = |\mathcal{M}|$ representing their count. The malicious users may return arbitrary output to perform a poisoning attack on $\mathcal{P}$. We refer to $\mathcal{H} = [n] \setminus \mathcal{M}$ as the set of honest users. We do not make any assumptions on how the malicious users are instantiated in practice – they could be fake accounts created by an adversary, or compromised real accounts, or a combination of both. Based on the practical implementation of the LDP protocol, there is an important distinction between the way in which the malicious users may carry out their poisoning attacks:

- **Input Poisoning.** Here the users do not have access to the implementation of the LDP randomizer. For instance, mobile applications might run proprietary code which the users do not have permission to tamper with. The only thing a malicious user can do is falsify their underlying input, i.e., change their input from $l_i$ to an arbitrary $l'_i$, and then report $q_i = R_i(l'_i)$ (Fig. 2a).
- **Response Poisoning.** This is a stronger threat model where a malicious user has direct control over the implementation of the LDP randomizer. For instance, the user could hack into the mobile application collecting their data. Consequently, the user can completely bypass the randomizer and submit an arbitrary response $q_i$ (Fig. 2b) to the aggregator.

Note that input poisoning applies to *any* protocol, private or not, because a user is free to change their input anytime. However, response poisoning attacks are unique to LDP – the distinction between

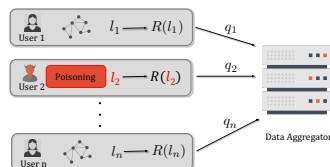
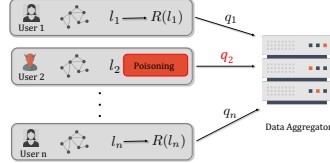

(a) Illustration of Input Poisoning         (b) Illustration of Response Poisoning

an user's input and their response is a characteristic of LDP which we results in a separation in the efficacy of the two types of attacks (see Sec. 6.3).

**Motivating Attacks.** Consider the following two realistic attacks for illustration. In a *degree inflation attack*, a target malicious user $U_t \in \mathcal{M}$ wants to inflate his degree estimate to affect his influence. He colludes with the other malicious users in $\mathcal{M}$ to do so, by requesting they behave as if they are connected with him. For example, if edges are being collected via $RR_\rho$, the responses of the malicious users will be biased towards reporting an edge (for reponse poisoning, these users may report a connection with no noise at all). Conversely, in a *degree deflation attack*, a target honest user $U_t \in \mathcal{H}$ is victimized by the malicious users in $\mathcal{M}$, who will behave as if they are not connected with $U_t$ even if they are. They may report that no edge exists between the users, and may deflate the degree estimate of $U_t$ or produce edge inconsistencies. Both of these attacks are rooted in reality and driven by monetary incentives, as it is common for brands to collaborate with popular users on social networks for marketing purposes ("influencers"). Moreover, these attacks are realistic, as it is easy for an adversary to create fake accounts and carry out a poisoning attack.

## 4 QUANTIFYING ROBUSTNESS

In this section, we present our formal framework for analyzing the robustness of a protocol for degree estimation. Specifically, we measure the robustness along two dimensions, *honest error* and *malicious error*. Intuitively, low honest error means that the protocol results in accurate degree estimates for honest users. On the other hand, low malicious error prevents a malicious user from manipulating their degree estimate too much *without* detection.

**Honest Error.** The *honest error* of a protocol quantifies the manipulation of an honest user's estimator. Specifically, malicious users can adversely affect an honest user $U_i \in \mathcal{H}$ by $(1)$ tampering with the value of $U_i$'s degree estimate $\hat{d}_i$ (by introducing additional loss in accuracy), or $(2)$ attempting to mislabel $U_i$ as malicious (by influencing the aggregator to report $\hat{d}_i = \perp$. The honest error of a protocol accounts for these scenarios and is formally defined as follows:

**Definition 2.** *(Honest Error) Let $\langle R_1, \ldots, R_n \rangle$ be a non-interactive, LDP protocol for degree estimation producing estimates $\langle \hat{d}_1, \ldots, \hat{d}_n \rangle$. Let $\mathcal{M}$ be a set of malicious users with $|\mathcal{M}| = m$ and $\mathcal{H}$ be the set of honest users. Then, the protocol has $(\alpha_1, \delta_1)$-honest error w.r.t. an attack from $\mathcal{M}$ if for all input graphs $G \in \mathcal{G}$ we have:*

$$\Pr\left[\forall U_i \in \mathcal{H}, \hat{d}_i = \perp \vee |\hat{d}_i - d_i| \geq \alpha_1\right] \leq \delta_1. \tag{1}$$

The parameter $\alpha_1$ dictates the utility of the estimate $\hat{d}_i$, and the parameter $\delta_1$ dictates the chance of failure—that either of the aforementioned conditions fail to hold. Thus, if a protocol has $(\alpha_1, \delta_1)$-honest error, it means that with probability at least $(1 - \delta_1)$, the degree estimate $\hat{d}_i$ for any honest user $U_i \in \mathcal{H}$ is inaccurate by *at most* $\alpha_1$, *and* $U_i$ is guaranteed to be *not* mislabeled as malicious. Lower the value of $\alpha_1$ and $\delta_1$, better is the robustness of the protocol for honest users.

**Malicious Error.** The *malicious error* of a protocol quantifies the manipulations of a malicious user's estimator. In particular, the protocol either returns a high accuracy estimate or returns $\hat{d}_i = \perp$ for these malicious users, regardless of the poisoning attack used. Formally, we use the following definition (which uses the complement event $\hat{d}_i \neq \perp \wedge |\hat{d}_i - d_i| \geq \alpha_2$):

**Definition 3.** *(Malicious Error) Let $\langle R_1, \ldots, R_n \rangle$ be a non-interactive, LDP protocol for degree estimation producing estimates $\hat{d}_1, \ldots, \hat{d}_n$. Let $\mathcal{M}$ be a set of malicious users with $|\mathcal{M}| = m$. Then, the protocol has $(\alpha_2, \delta_2)$-malicious error w.r.t an attack from $\mathcal{M}$ if, for all input graphs $G \in \mathcal{G}$:*

$$\Pr\left[\forall U_i \in \mathcal{M}, \hat{d}_i \neq \perp \wedge |\hat{d}_i - d_i| \geq \alpha_2\right] \leq \delta_2. \tag{2}$$

Like with honest error, the parameter $\alpha_2$ dictates the accuracy of the estimate $\hat{d}_i$, and $\delta_2$ dictates the chance of failure. As an important distinction, note that the failure event is when $\hat{d}_i$ is both $\neq \perp$ *and* has poor utility, as stated above. Thus, the $\vee$ used in the definition of honest error is replaced by a $\wedge$. In other words, a protocol has $(\alpha_2, \delta_2)$-malicious error if for any malicious user $U_i, i \in \mathcal{M}$, the protocol (1) fails to identify $U_i$ as malicious, *and* (2) reports its degree estimate $\hat{d}_i$ with an accuracy loss *greater* than $\alpha_2$, with probability at most $\delta_2$. Lower the value of $\alpha_2$ and lower the value of $\delta_2$, better is the robustness.

> **Note.** Our proposed framework provides a strong notion of robustness – not only are we able to guarantee high utility estimates, but also detect *and* flag individual malicious users (by reporting $\perp$). The assumption of bounding the number of adversaries ($m$) is inspired by standard assumptions in cryptography Goldreich (2001) and erasures in coding theory cod. In practice, $m$ can be set based on empirical evidence Shejwalkar et al. (2022).

## 5 ROBUSTNESS LOWER BOUNDS FOR LDP PROTOCOLS

Here, we present a lower bound on the error of LDP protocol under poisoning. To do this, we simplify the problem of degree vector estimation to a simple task of distinguishing two scenarios, and then appeal to information-theoretic lower bounds in LDP. In our first scenario, consider an "honest world" where user $U_n$ follows the protocol honestly, but the other malicious users manipulate their inputs to erase any edge to $U_n$. In the second scenario, or the "malicious world", user $U_n$ behaves maliciously and inflates his degree (using input poisoning) by about $\frac{m}{4} + \frac{\sqrt{n}}{40\epsilon}$ such that it matches the degree of $U_n$ in the honest world, and the malicious users manipulate their inputs to accordingly report an edge. The key idea of the attack in the malicious world is to design the input poisoning of $U_n$ such that his output is identical to what it would be in the honest world. Thus, the only feature left to distinguish the two worlds are the responses from user $U_1, \ldots, U_{n-1}$, which are themselves subject to information-theoretic lower bounds on LDP Duchi et al. (2013). This is the key to obtaining the $\frac{\sqrt{n}}{\epsilon}$ term in the error bound—if the output of $U_n$ in the malicious world were not crafted to be indistinguishable from the output in the honest world, the privacy error term would be $\frac{1}{\epsilon}$, which is the typical error term in the central model of DP. Formally, our lower bound is:

**Theorem 1.** *For any $m, n$, and $\epsilon < \frac{1}{20}$, and any non-interactive, $\epsilon$-LDP protocol, there is an input manipulation attack such that the protocol has either $(\frac{m}{4} + \frac{\sqrt{n}}{40\epsilon}, 0.1)$-honest error or $(\frac{m}{4} + \frac{\sqrt{n}}{40\epsilon}, 0.1)$-malicious error w.r.t the attack.*

Since input poisoning is a subset of response poisoning, this lower bound applies to both types of protocols. Our lower bound applies to all $\epsilon < \frac{1}{20}$, and typically small values of $\epsilon$ are of interest (corresponding to high privacy). With more careful bookkeeping of constants, the maximum value of $\epsilon$ can almost certainly be increased. While we conjecture that a tighter lower bound for response poisoning is $\Omega(m + \frac{m}{\epsilon} + \frac{\sqrt{n}}{\epsilon})$, proving this fact is more difficult and we leave it as future work.

Now, we demonstrate that naive, baseline protocols fall far from these lower bounds. We summarize our discussion here, and defer a formal discussion to App. A.

**Laplace Mechanism.** The simplest mechanism for estimating degree is the Laplace mechanism, $R_{Lap}$, where each user directly reports their degree estimates plus $Lap(\frac{\log(1/\delta)}{\epsilon})$ noise. Consequently, the degree estimate of an honest user *cannot* be tampered with at all, and each user attains just $\log \frac{\log(1/\delta)}{\epsilon}$ error per user. Thus, the Laplace mechanism has $O(\frac{\log(1/\delta)}{\epsilon}, \delta)$-honest error, and the per-user loss in accuracy is comparable with the Laplace mechanism in central DP. On the flip side, a malicious user can flagrantly lie about their estimate without detection, meaning the protocol has *not* $(\alpha, \delta)$-malicious error for any $\alpha < n - 1$ and $\delta < 1$. In other words, there exists a graph and an attack against $R_{Lap}$ in which a malicious user is *guaranteed* to manipulate their true degree by $n - 1$.
**Randomized Response.** Now, consider the mechanism where users release their edges via randomized response. As each edge is shared by two users, edge $(i, j)$ is reported by just one of the users based on their index. We will refer to this approach as *SimpleRR* (Alg. 2). The aggregator counts the total number of edges to user and then debias the estimate of the degree. Since up to $m$ of a user's edges may be reported by malicious user, we can show this protocol has $(m + \frac{m + \sqrt{n \log(1/\delta)}}{\epsilon}, \delta)$-

honest error (with the $\sqrt{n}$ term coming from the error of randomized response). However, since a malicious user may in the worst case fabricate *all* of their edges, this protocol has again *not* $(\alpha, \delta)$-malicious error for any $\alpha < n - 1$ and $\delta < 0$. In other words, a malicious user can always get away with the worst-case $n - 1$ manipulation. A detailed discussion is in App. A.

# 6 PROPOSED ROBUST PROTOCOLS

In this section, we present our proposed protocols for robust degree estimation. We start by focusing on response poisoning and design protocols that improve both the malicious error and the honest error. We conclude by discussing our results for input poisoning which are optimal, i.e., matches the lower bound of Thm. 1 (up to constants).

## 6.1 IMPROVING MALICIOUS ERROR WITH VERIFICATION

***RRCheck* Protocol.** As discussed earlier, the naive *SimpleRR* protocol has high malicious error. To tackle this, we propose a new protocol, *RRCheck*(Alg. 1), as follows. *RRCheck* enhances the data collected by *SimpleRR* with verification – for edge $e_{ij} \in E$, instead of collecting a noisy response from just one of the users $U_i$ or $U_j$, *RRCheck* collects a noisy response from *both* users. This creates data redundancy which can then be checked for consistency. Specifically, the estimator counts only those edges $e_{ij}$ for which *both* $U_i$ and $U_j$ are consistent and report a 1. The count of noisy edges involving user $U_i$ is then given by:

$$count_i^{11} = \sum_{j \in [n] \setminus i} q_i[j] q_j[i].$$

The unbiased degree estimate of $U_i$ is computed as follows:

$$\hat{d}_i = \frac{count_i^{11} - \rho^2(n-1)}{1 - 2\rho}. \tag{3}$$

---

**Algorithm 1** *RRCheck*: $\{0,1\}^{n \times n} \mapsto \{\mathbb{N} \cup \{\perp\}\}^n$

    **Parameters:** $\epsilon$ - Privacy parameter;
    $\tau$ - Threshold for consistency check;
1: $\rho = \frac{1}{1+e^\epsilon}$
2: **for** $i \in [n]$ **do**
3:     $q_i = RR_\rho(l_i)$
4: **for** $i \in [n]$ **do**
5:     $count_i^{11} = \sum_{j \in [n] \setminus i} q_i[j] q_j[i]$
6:     $count_i^{01} = \sum_{j \in [n] \setminus i} (1 - q_i[j]) q_j[i]$
7:     **if** $|count_i^{01} - \rho(1-\rho)(n-1)| \leq \tau$ **then**
8:         $\hat{d}_i = \frac{1}{1-2\rho}(count_i^{11} - \rho^2(n-1))$
9:     **else**
10:         $\hat{d}_i = \perp$
    **return** $(\hat{d}_1, \hat{d}_2, \ldots, \hat{d}_n)$

---

For robustness, *RRCheck* imposes a check on the number of instances of inconsistent reporting ($U_i$ and $U_j$ differ in their respective bits reported for their mutual edge $e_{ij}$). For every user $U_i$, the protocol has an additional capability of returning $\perp$ whenever the consistency check fails, indicating that the aggregator believes that $U_i$ is malicious. The intuition is that if the $U_i$ is malicious and attempts to poison a lot of the edges, then there would be a large number of inconsistent reports for the edges to honest users. *RRCheck* counts the number of inconsistent reports for $U_i$ as:

$$count_i^{01} = \sum_{j=1}^{n} (1 - q_i[j]) q_j[i],$$

i.e., the number of edges connected to user $U_i$for which they reported 0 and user $U_j$ reported 1. Intuitively, the check computes the expected number of inconsistent reports assuming $U_i$ to be honest and flags $U_i$ in case the reported number is outside a confidence interval. Formally, if

$$|count_i^{01} - \rho(1-\rho)(n-1)| \leq \tau, \tag{4}$$

then set $\hat{d}_i = \perp$, where $\tau = m + \sqrt{3n\rho \ln \frac{2}{\delta}}$ is a threshold. This check forces a malicious user to send a response with only a small number of poisoned edges (as allowed by the threshold $\tau$), thereby significantly restricting the attack strength. For example, they are not able to indicate they are connected to all users in the graph, as this would produce a large number of inconsistent edges.

Note that due to the randomization required for LDP, some honest users might also fail the check. However, we observe that for two honest users $U_i$ and $U_j$, the product term $(1 - q_i[j]) q_j[i]$ follows the Bernoulli$(\rho(1-\rho))$ distribution, irrespective of whether the edge $e_{ij}$ exists. Consequently $count_i^{01}$ is tightly concentrated around its mean. This ensures that the probability of mislabeling an honest user (by returning $\perp$) is low.

**Theorem 2.** *Let $\mathcal{M}$ be a set of malicious users with $|\mathcal{M}| = m$. Then, the RRCheck protocol run with threshold $\tau = m + \sqrt{2\rho n \ln \frac{4n}{\delta}}$ has $\left(2m(\frac{e^\epsilon + 1}{e^\epsilon - 1}) + 4\sqrt{n} \frac{\sqrt{(e^\epsilon + 1)\ln(4n/\delta))}}{e^\epsilon - 1}, \delta\right)$-honest and malicious error w.r.t any response poisoning from $\mathcal{M}$ (proof is presented in App. G.5).*

The additional verification of *RRCheck* results in a clear improvement — the malicious users can now skew their degree estimates only by a limited amount (as determined by the threshold $\tau$) or risk getting detected, which results in a lower malicious error. Specifically, a malicious user can now only skew their degree estimate by at most $\tilde{O}\left(m(1 + \frac{1}{\epsilon}) + \frac{\sqrt{n}}{\epsilon}\right)$ for response poisoning attacks, respectively (as compared to $n - 1$ in Thm. 7). Intuitively, the $\frac{\sqrt{n}}{\epsilon}$ term comes from the error introduced by randomized response. The $m(1 + \frac{1}{\epsilon})$ term comes from the adversarial behavior of the malicious users – $m$ term is inevitable and accounts for the worst case scenario where all $m$ malicious users are colluding and report consistently, while the $\frac{1}{\epsilon}$ factor corresponds to the scaling factor required for de-biasing. This observation is in line with prior work Cheu et al. (2021) that assesses the impact of poisoning attacks on tabular data.

The robustness guarantees in the above theorem worsen with smaller $\epsilon$. This is because at lower privacy, the collected responses are more noisy thereby making it harder to distinguish honest users from malicious ones. In particular, a protocol should not return $\perp$ for honest users (i.e., mislabel them) to ensure good robustness. Here, more malicious error is tolerated before a $\perp$ is returned for a malicious user. This is evident in Eq. 4–threshold $\tau$ grows with smaller $\epsilon$. See App. B for more details on this price of privacy.

Interestingly for response poisoning, the degree deflation attack (Sec. 3) is a worst-case attack for honest error – the attack can skew an honest user's degree estimate by $\Omega\left(m(1 + \frac{1}{\epsilon}) + \frac{\sqrt{n}}{\epsilon}\right)$. Similarly, the degree inflation attack can skew a malicious user's degree estimate by $\Omega\left(m(1 + \frac{1}{\epsilon}) + \frac{\sqrt{n}}{\epsilon}\right)$) resulting in the worst-case malicious error.

> **Note.** Our robustness results (including the ones presented later) are completely *attack-agnostic* – they hold for any attack, for any number of malicious users $m$, and all graphs.

### 6.2 Improving Honest Error with a Hybrid Protocol

The robustness guarantees for *RRCheck* contain a $\tilde{O}(\frac{\sqrt{n}}{\epsilon})$ term coming from the noise in randomized response. This is inherent in *any* randomized response based mechanism Beimel et al. (2008); Chan et al. (2012) since each of the $n$ bits of the adjacency list need to be independently randomized. On the other hand, $R_{Lap}$ provides a more accurate degree estimate for the honest users but has the worst-case $(n - 1)$ malicious error (see Thm. 6). In this section, we present a mitigation strategy. The key idea is to combine the two approaches and use a hybrid protocol, *Hybrid*, that achieves the best of both worlds – honest error of $R_{Lap}$, and malicious error of *RRCheck*.

The *Hybrid* protocol is outlined in Alg. 4 and described as follows. Each user $U_i$ prepares two responses – the noisy adjacency list, $q_i$, randomized via $\mathsf{RR}_\rho$, and a noisy degree estimate, $\tilde{d}_i^{lap}$, perturbed via $R_{Lap}$, and sends them to the data aggregator. $U_i$ divides the privacy budget between the two responses according to some constant $c \in (0, 1)$. The data aggregator first processes each list $q_i$ to employ the same consistency check on $count_i^{01}$ as that of the *RRCheck* protocol (Step 9). In case the check passes, the aggregator computes the unbiased degree estimate $\tilde{d}_i^{rr}$ from $count_i^{11}$, in the exact same way as *RRCheck*. Note that $\tilde{d}_i^{rr}$ and $\tilde{d}_i^{lap}$ are the noisy estimates of the *same* ground truth degree, $d_i$, computed via two different randomization mechanisms. To this end, the aggregator employs a second check (Step 11) to verify the consistency of the two estimates:

$$|\tilde{d}_i^{rr} - \tilde{d}_i^{lap}| \leq \frac{2\tau}{1 - 2\rho} + \frac{1}{(1-c)\epsilon} \ln \frac{2n}{\delta},$$

where $\rho$ in this case is equal to $\frac{1}{1 + e^{c\epsilon}}$. This check accounts for the noise from $\tilde{d}_i^{rr}$ (the $\frac{2\tau}{1 - 2\rho}$) term, and the noise from $\tilde{d}_i^{lap}$ (the $\frac{1}{(1-c)\epsilon} \ln \frac{2n}{\delta}$ term). Finally, the protocol returns $\perp$ if either of the checks fail. In the event that both the checks pass, the aggregator uses $\tilde{d}_i^{lap}$ (obtained via $R_{Lap}$) as the final degree estimate $\hat{d}_i$ for $U_i$.

Each $\hat{d}_i^{rr}$ estimate is computed identically to that of *RRCheck*. *Hybrid* allows a user to send an

even more accurate estimate of their degree – to prevent malicious users from outright lying about this value, $\hat{d}_i^{lap}$ is compared to $\hat{d}_i^{rr}$. This allows *Hybrid* to enjoy the honest error of $R_{Lap}$ and the malicious error of *RRCheck*. Formally,

**Theorem 3.** $\mathcal{M}$ *is a set of* $m$ *malicious users. For all* $c \in (0,1)$*, the Hybrid protocol run with* $\tau = m + \sqrt{2\rho n \ln \frac{8n}{\delta}}$ *has* $(\frac{\ln(2n/\delta))}{(1-c)\epsilon}, \delta)$*-honest error and* $\left( 4m(\frac{e^{c\epsilon}+1}{e^{c\epsilon}-1}) + 8\sqrt{n}\frac{\sqrt{(e^{c\epsilon}+1)\ln(8n/\delta)}}{e^{c\epsilon}-1} + \frac{\ln(2n/\delta))}{(1-c)\epsilon}, \delta \right)$*- malicious error w.r.t any response poisoning attack from* $\mathcal{M}$ *(The proof is in App. G.7.).*

We remark that *Hybrid* achieves the optimal honest error of $\tilde{O}(\frac{1}{\epsilon})$ that is achievable under LDP. This is due to the fact that the data aggregator uses $\tilde{d}_i^{lap}$ as its final degree estimate. The malicious error can be written as $\tilde{O}(m(1 + \frac{1}{\epsilon}) + \frac{\sqrt{n}}{\epsilon})$ which is the same as that of *RRCheck*. This is enforced by the two consistency checks. Hence, the hybrid mechanism achieves the best of both worlds. Observe that the above result does not violate the lower bound –while that protocol achieves $O(\frac{1}{\epsilon})$ honest error, its malicious error is higher than the bound stated in Thm. 1.

### 6.3   OPTIMAL RESULTS FOR INPUT POISONING ATTACKS

So far we have only considered response poisoning attacks where the malicious users are free to report arbitrary responses to the aggregator. However, to carry out such an attack, a user would have to bypass the LDP data collection mechanism's security features, such as those in mobile applications preventing unauthorized code tampering. Given its very realistic practical threat, here we study the impact of input poisoning attacks. Note that input poisoning attacks are strictly weaker than response poisoning attacks. This is because the poisoned input is randomized to satisfy LDP in the former which introduces noise in the final output, thereby weakening the adversary's signal. Hence intuitively, we hope to obtain better robustness against input poisoning attacks.

First, we show the result for *SimpleRR*. There is an improvement in both the honest error and the malicious error, because the adversary's signals in the poisoned data (such as, a malicious user indicating they share an edge with every other user, or $m$ malicious users intentionally deleting their edges to an honest user), are noised via randomized response which weakens them.

**Theorem 4.** *Let* $\mathcal{M}$ *be a set of* $m$ *malicious users. SimpleRR has* $(m + \sqrt{n}\frac{\sqrt{2(e^\epsilon+1)\ln\frac{4n}{\delta}}}{e^\epsilon-1}, \delta)$*-honest error and* $(n-1, \frac{1}{2})$*-malicious error w.r.t any input poisoning from* $\mathcal{M}$ *(proof in App. G.9).*

Written asymptotically, the honest error of Thm. 4 is $\tilde{O}(m + \frac{\sqrt{n}}{\epsilon})$, which improves the guarantee over response poisoning attacks (Thm. 7) by a factor of $\frac{m}{\epsilon}$. This shows a separation between input and response poisoning attacks. A similar case holds for the malicious error – while a malicious user can *always* manipulate their degree by $n-1$ under response poisoning attacks, for input poisoning attacks, *SimpleRR* has $(n-1, \frac{1}{2})$-malicious error.

Despite exhibiting improvement over response poisoning, the naive protocols still fall short of providing acceptable malicious error. Here, we analyze the robustness of our proposed protocols, *SimpleRR* and *Hybrid*, under input poisoning. For both the mechanisms here, we can set a smaller value for $\tau$, the threshold for checking the number of inconsistent edges. This is because the number of inconsistent edges is more concentrated around its means, and hence, a tighter confidence interval with a smaller $\tau$ suffices. Thus, both the honest error and the malicious error of the protocols are improved. For lack of space, the results for *SimpleRR* are in App. C. For *Hybrid*, we have:

**Theorem 5.** *For any* $c \in (0,1)$*, the Hybrid protocol with threshold* $\tau = m(1-2\rho) + \sqrt{8\max\{\rho n, m\}\ln\frac{8n}{\delta}}$ *has* $(\frac{1}{(1-c)\epsilon}\ln\frac{4n}{\delta}, \delta)$*-honest error and* $(4m + 8\sqrt{\max\{n, m(e^{c\epsilon}+1)\}}\frac{\sqrt{2(e^{c\epsilon}-1)\ln\frac{8n}{\delta}}}{e^{c\epsilon}+1}, \delta)$*-malicious error w.r.t any input poisoning for any set of* $m$ *malicious users (proof in App. G.11).*

Written asymptotically, the honest error of *Hybrid* is $(\tilde{O}(\frac{1}{\epsilon}), \delta)$, and its malicious error is $(\tilde{O}(m + \frac{\sqrt{n}}{\epsilon}), \delta)$. Compared with Thm. 3 for response poisoning, *Hybrid* has similar honest error since the data aggregator uses the degree estimate collected via $R_{Lap}$ as its final estimate as before. However, the malicious error is improved by an additive factor of $O(\frac{m}{\epsilon})$, which comes from the smaller $\tau$.

> **Note.** The robustness results for *Hybrid* for input poisoning are order optimal, i.e., they match the lower bound in Thm. 1 (up to constant factors).

# 7 EVALUATION

In this section, we answer the following questions:

- **Q1.** How do the different protocols perform in terms of the honest error and the malicious error?
- **Q2.** How do the efficacies of input and response poisoning attacks compare?

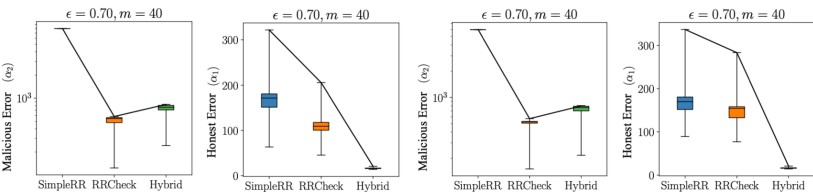

(a) *FB:* Degree Inflation    (b) *FB:* Degree Deflation    (c) *Syn:* Degree Inflation    (d) *Syn:* Degree Deflation

Figure 3: Robustness Analysis: The whiskers range from the maximum to the minimum empirical honest error and malicious error observed across all the attacks of the specific type. The line corresponds to the strongest evaluated attack of the specific type.

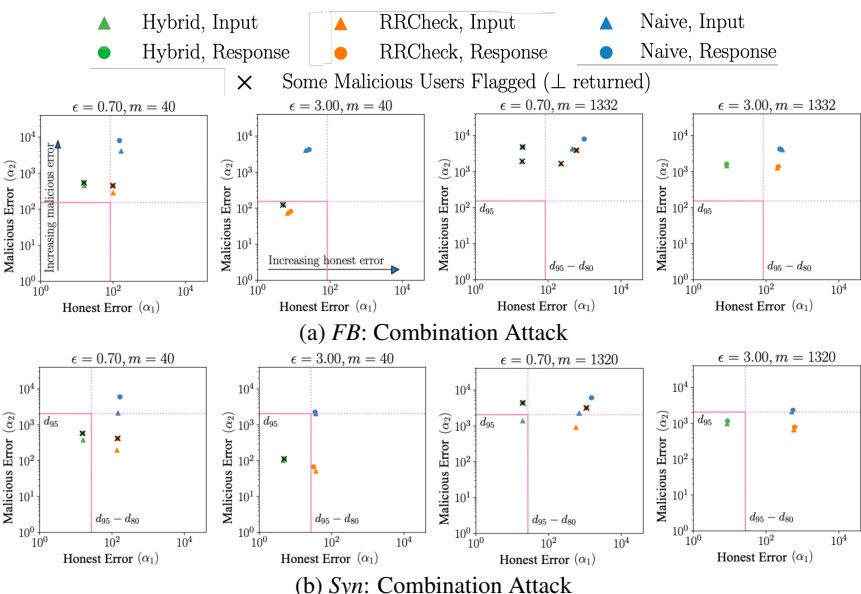

Figure 4: Comparison of input and response poisoning: We plot the empirical honest error and malicious error. $d_k$ denotes the degree of the $k$ percentile node.

**Experimental Setup.** We consider two graphs – a real-world sparse graph, *FB* and a synthetically generated dense graph, *Syn*. Due to lack of space, more details about the experimental setup are in App. D. We carry out an extensive analysis of the robustness of our protocols by evaluating against **16** different attacks. The attacks are broadly classified into two types – degree inflation and degree deflation where the goal of the malicious users is to increase (resp. decrease) the degree estimate of a target malicious (resp. honest) user by as much as possible. We choose these attacks because firstly, they can meet the asymptotic theoretical error bounds. Secondly, these attacks are grounded in real-world motivations and represent practical threats (Sec. 3). The 16 attacks evaluated represent different configurations of the degree inflation and deflation attacks. Specifically, they differ in $(1)$ the number of targets (both malicious or honest users), $(2)$ how the non-target malicious users are chosen $(3)$ collusion strategies. The different configurations capture a multitude real-world attack scenarios and adversarial goals (Tab. 2 in App. E). Due to the lack of space, we defer the details of all the attacks in App. E. For every attack we report the maximum loss in accuracy over all the honest targets (honest error, $\alpha_1$) and the malicious targets (malicious error, $\alpha_2$). We run each experiment 50 times and report the mean. We use $\delta = 10^{-6}$ and $c = 0.9$ for *Hybrid*. Extra results are in App. D.2.

## 7.1 ROBUSTNESS ANALYSIS

We focus on response poisoning to analyze the worst-case scenario (since response poisoning is stronger than input poisoning). The number of malicious users is set to $m = 1\%$ and $\epsilon = 0.7$.

**Degree Inflation.** We report the empirical malicious error for all the attacks that have a degree inflation component (10 out of the 16 attacks) in Figs. 3a and 3c for *FB* and *Syn*, respectively. In other words, these are all the attacks where at least a subset of the malicious users are trying to inflate the degree of some (malicious) target. We observe that both our proposed protocols perform significantly better than the baseline. For instance, for the strongest inflation attack we evaluated (attack A11 in Tab. 2) – *SimpleRR* has $9.7\times$ and $13.8\times$ higher malicious error than *Hybrid* and *RRCheck*, respectively, for *FB*. Note that *RRCheck* performs slightly better than *Hybrid*. This is because, although both the protocols have the same asymptotic malicious error, the constants are higher for *Hybrid* (see Sec. 6.2). Finally, for both datasets our protocols flag the malicious targets (returning $\perp$) for $51\% - 70\%$ of the trials (Tab. 1). This indicates that our proposed protocols are able to detect malicious users, thereby disincentivizing malicious activity.

**Degree Deflation.** Figs. 3b and 3d show the results for the degree deflation attacks on *FB* and *Syn*, respectively. Specifically, we report the empirical honest error for all the attacks that have a (11 out of the 16 attacks evaluated). We observe that *Hybrid* performs the best. For instance, for the strongest deflation attack we evaluated (attack A8 in Tab. 2) it has $16.2\times$ and $13.6\times$ lower honest error than *SimpleRR* and *RRCheck*, respectively, for *Syn*. Additionally, our protocols are able to flag malicious users when they target a large number of honest users. Specifically, for the strongest degree deflation attack, *Hybrid* flags $4.5\%$ and $49.8\%$ of the malicious users for *FB* and *Syn*, respectively. *RRCheck*, on the other hand, flags $3\%$ and $59.3\%$ of the malicious users for *FB* and *Syn*, respectively. Note that the number of actual honest users affected by a malicious user is bounded by its degree. Hence, the rate of flagging is less aggressive for *FB* since it is a sparse graph.

## 7.2 COMPARING INPUT AND RESPONSE POISONING

We plot the efficacies of input and response poisoning attacks in Fig. 4. For this, we choose the strongest combination attack we evaluated (A10 in Tab. 2). We observe that input poisoning is weaker than response poisoning in terms of both the honest error and the malicious error. Specifically, the malicious error is worse than response poisoning for all three protocols (since response poisoning has an extra $O(\frac{m}{\epsilon})$ term). In terms of the honest error, input poisoning is worse than response poisoning (again because of the extra $O(\frac{m}{\epsilon})$ term in response poisoning). $m = 33\%, \epsilon = 0.7$. However, the honest error for *Hybrid* is comparable for both input and response poisoning which is consistent with our theoretical results (Thm. 5). This is because under both types of attacks, *Hybrid* uses the honest users' Laplace estimates which are not affected by the malicious users. As expected, the separation between input and response poisoning becomes less prominent with higher $\epsilon$ and lower $m$, as it is harder to pull off strong attacks for these regimes.

We also mark the degree of $95^{th}$ percentile node ($d_{95}$) for the graphs in the plots. The way to interpret this is as follows. If an error of a protocol falls below the line, then any malicious user can inflate their degree estimate to be in the $> d_{95}$ percentile by staging a poisoning attack. Our protocols perform better for the dense graph *Syn* (attacks are prevented for both values of $\epsilon$). This is because of the $\tilde{O}(\frac{\sqrt{n}}{\epsilon})$ term in the malicious error – this term dominates the malicious error for sparse graphs.

We plot the measure $d_{95} - d_{80}$ in Figs. 4a and 4b where $d_k$ denotes the degree of the $k$ percentile node. The way to interpret this is as follows. If an error falls to the left of the line, then malicious users can successfully deflate the degree of an honest target from $> 95$ percentile to $< 80$ percentile. Based on our results, we observe that *Hybrid* is mostly effective in protecting against this attack even with a large number of malicious users of $m = 33\%$.

## 8 CONCLUSION

In this paper, we have investigated the impact of poisoning attacks on degree vector estimation for graphs under LDP. We have presented a formal framework for analyzing the robustness of a protocol against poisoning. Our framework can quantify the impact of a poisoning attack on both honest and malicious users. We have shown a lower bound on such poisoning attacks. Additionally, we have proposed novel robust degree estimation protocols under LDP by leveraging the natural data redundancy in graphs that can match the lower bound for a specific class of attacks.

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

## A  QUANTIFYING ROBUSTNESS CNTD.

Complementary to Defn. 2 in Sec. 4, we introduce the notion of $\alpha_1$-*tight honest error*. A protocol has $\alpha_1$-tight honest error w.r.t an attack, if there exists a graph such that the attack is *guaranteed* to either skew the the degree estimate of at least one honest user by *exactly* $\alpha_1$. We use this definition to show the existence of strong attacks that are guaranteed to be very successful in manipulating the data of an honest user, which motivates the need for robust solutions.

Similarly for malicious error we introduce the notion of $\alpha_1$-*malicious error*. A protocol has $\alpha_1$-tight malicious error w.r.t an attack if there exists a graph $G \in \mathcal{G}$ such that at least one malicious user is *guaranteed* to have their degree estimate mis-estimated by *exactly* $\alpha_1$ *without* getting detected. In other words, an attack with $(n-1)$-tight honest error/malicious error represents the strongest possible attack– an user's estimate can *always* be skewed by the worst-case amount. We use this definition to motivate the need for more robust protocols.

### IMPACT OF POISONING ON BASELINE MECHANISMS CNTD.

Within our robustness framework, we analyze the two naive private mechanisms – the Laplace mechanism and randomized response. The shortcomings of these mechanisms motivate the design of our robust protocols as discussed in the main paper.

### A.1  LAPLACE MECHANISM

The simplest mechanism for estimating an user's degree is the Laplace mechanism, $R_{Lap}$, where each user directly reports their noisy estimates. Consequently, the degree estimate of an honest user *cannot* be tampered with at all – the $\tilde{O}(\frac{1}{\epsilon})$[1] term is due to the error of the added Laplace noise. This error is in fact optimal (matches that of central DP) for degree estimation. On the flip side, a malicious user can flagrantly lie about their estimate without detection resulting in the the worst-case malicious error. Specifically, there exists a graph and an attack against $R_{Lap}$ in which a malicious user is guaranteed to manipulate their true degree by $n-1$ – this holds for the case where the malicious user is an isolated node but lies that their degree is $n-1$. The robustness of $R_{Lap}$ against response poisoning attacks is formalized as follows:

**Theorem 6.** *Let $\mathcal{M}$ be a set of malicious users with $|\mathcal{M}| = m$. The $R_{Lap}$ protocol has $(\frac{1}{\epsilon} \log \frac{n}{\delta}), \delta)$- honest error w.r.t any response poisoning from $\mathcal{M}$. However, there is a response poisoning attack $\mathcal{A}$ such that $R_{Lap}$ has $(n-1)$-tight malicious error w.r.t $\mathcal{A}$.*

The proof is in App. G.3. Thus according to our robustness framework, $R_{Lap}$ has low honest error but high malicious error. Intuitively, $R_{Lap}$ fails to provide low malicious error because there is no way to verify the malicious users' reports. It is important to note that $R_{Lap}$ has low honest error even with $n-1$ malicious users while the worst-case malicious error is inevitable even with a single malicious user.

### A.2  RANDOMIZED RESPONSE

In this section, we look at an alternative mechanism where the users release their edges via randomized response. Recall that the information about an edge is shared between two users – the idea here is to leverage this *distributed information*. For our baseline algorithm, *SimpleRR* (described in Alg. 2), the data aggregator collects information about an edge from a *single* user. Specifically, for edge $(i, j)$

---

[1]$\tilde{O}$ hides factors of $\log \frac{1}{\delta}$

with $i < j$, it simply uses the response from user $U_i$ to decide if the edge exists. To estimate the degree, it counts the total number of edges to user $U_i$ with the random variable $count_i^1$ and then computes a debiased estimate of the degree. Note that this naive approach is used by many prior works in graph algorithms Wang et al. (2016); Qin et al. (2017a); Imola et al. (2021; 2022). Formally

---

**Algorithm 2** *SimpleRR*: $\{0,1\}^{n\times n} \mapsto \{\mathbb{N} \cup \{\perp\}\}^n$

---

    **Parameter:** $\epsilon$ - Privacy parameter
    **Input:** $\{l_1, \cdots, l_n\}$ where $l_i \in \{0,1\}^n$ is $U_i$'s adjacency list;
    **Output:** $(\hat{d}_1, \cdots, \hat{d}_n)$ where $\hat{d}_i$ is $U_i$'s degree estimate;
    **Users**
1: **for** $i \in [n]$ **do**
2:     $q_i = RR_\rho(l_i)$
    **Data Aggregator**
3: **for** $i \in [n]$ **do**
4:     $count_i^1 = \sum_{j<i} q_j[i] + \sum_{i<j} q_i[j]$
5:     $\hat{d}_i = \frac{1}{1-2\rho}(count_i^1 - \rho(n-1))$
    **return** $(\hat{d}_1, \hat{d}_2, \ldots, \hat{d}_n)$

---

for response poisoning attacks, we have:

**Theorem 7.** *Let $\mathcal{M}$ be a set of malicious users with $|\mathcal{M}| = m$. Then, the SimpleRR protocol has $(m\frac{e^\epsilon+1}{e^\epsilon-1} + \sqrt{n}\frac{\sqrt{(e^\epsilon+1)\ln\frac{2n}{\delta}}}{e^\epsilon-1}, \delta)$-honest error w.r.t any response poisoning attack from $\mathcal{M}$. However, there is a response poisoning attack $\mathcal{A}$ such that SimpleRR has $(n-1)$-tight malicious error with respect to $\mathcal{A}$.*

The above theorem is proved in App. G.4. For $\epsilon < 1$, the honest error is $\approx m(1+\frac{1}{\epsilon}) + \frac{\sqrt{n}}{\epsilon}$. Intuitively, the $\frac{\sqrt{n}}{\epsilon}$ term comes from the error introduced by randomized response. The $m(1+\frac{1}{\epsilon})$ term comes from the adversarial behavior of the malicious users – $m$ term is inevitable and accounts for the worst case scenario where all $m$ malicious users are colluding (see discussion at the end of Sec. 6.3), while the $\frac{1}{\epsilon}$ factor corresponds to the scaling factor required for de-biasing. This observation is in line with prior work Cheu et al. (2021) that assesses the impact of poisoning attacks on tabular data. Clearly, smaller the value of $\epsilon$, worse is the attack's impact.

Similar to the Laplace mechanism, *SimpleRR* has $(n-1)$-tight malicious error, i.e., a malicious user can always get away with the worst-case $n-1$ error. This happens when $U_n$ is an isolated node who acts maliciously and reports an all-one list. Thus, once again this worst-case malicious error is inevitable even with a single malicious user.

## B  PRICE OF PRIVACY

The randomization required to achieve privacy adversely impacts a protocol's robustness to poisoning. Here, we perform an ablation study, and formalize the price of privacy by comparing to the honest error and malicious error of *non-private* protocols. For this, we adapt our consistency check to the non-private setting via the *DegCheck* protocol (Alg. 3) as described below. First, every user reports their true adjacency list to the data aggregator. The data aggregator then employs a consistency check to identify the malicious users. Due to the absence of randomization, the check is much simpler here and involves just ensuring that the number of inconsistent reports for user $U_i$ is bounded by $m$, i.e., $count_i^{01} + count_i^{10} \leq m$. In case the check goes through, the aggregator can directly use $count_i^{11}$, the count of the edges where both users have reported 1s consistently, as the degree estimate $\hat{d}_i$. We quantify the impact of the poisoning attacks on *DegCheck* as follows.

**Theorem 8.** *Let $\mathcal{M}$ be a set of malicious users with $|\mathcal{M}| = m$. Then, there are poisoning attacks $\mathcal{A}_1$ and $\mathcal{A}_2$ such that the DegCheck protocol has $m$-tight honest error w.r.t $\mathcal{A}_1$ and $(\min\{2m-1, n-1\})$-tight malicious error w.r.t $\mathcal{A}_2$.*

The proof of the above theorem is in App. G.6. Note that the robustness guarantees are tight in that there are attacks which always successfully attain $m$ error for an honest user and $\min\{2m-1, n-1\}$

---

**Algorithm 3** *DegCheck*: $\{0,1\}^{n \times n} \mapsto \{\mathbb{N} \cup \{\bot\}\}^n$

---

    **Parameter:** $m$ - Number of malicious users;
    **Input:** $\{l_1, \cdots, l_n\}$ where $l_i \in \{0,1\}^n$ is $U_i$'s adjacency list;
    **Output:** $(\hat{d}_1, \cdots, \hat{d}_n)$ where $\hat{d}_i$ is $U_i$'s degree estimate;
    **Users**
1: **for** $i \in [n]$ **do**
2:     $q_i = l_i$
    **Data Aggregator**
3: **for** $i \in [n]$ **do**
4:     $count_i^{11} = \sum_{j \in [n] \backslash i} q_i[j] q_j[i]$
5:     $count_i^{01} = \sum_{j \in [n] \backslash i} (1 - q_i[j]) q_j[i]$
6:     $count_i^{10} = \sum_{j \in [n] \backslash i} q_i[j] (1 - q_j[i])$
7:     **if** $(count_i^{01} + count_i^{10}) \leq m$ **then**
8:         $\hat{d}_i = count_i^{11}$
9:     **else**
10:         $\hat{d}_i = \bot$
    **return** $(\hat{d}_1, \hat{d}_2, \ldots, \hat{d}_n)$

---

for a malicious user. Thus, the low-order manipulation term of $O(m)$ is inevitable even for non-private protocols based on consistency checks.

Comparing Thm. 8 to Thm. 2, we see an improvement in both the honest error and malicious error guarantees over the private protocols – the malicious users can skew the degree estimates by only $O(m)$, and the $\tilde{O}(\frac{m}{\epsilon} + \frac{\sqrt{n}}{\epsilon})$ terms have disappeared. This highlights the price of privacy – the private protocols incur additional error due to the randomization of LDP.

Thus, our proposed *RRCheck* protocol shows that the malicious error of a degree estimation protocol can be significantly improved by leveraging the redundancy in graph data. Additionally, we observe the robustness of the protocol worsens with higher privacy and we explicitly formalize price of privacy.

## C    Input Poisoning Attack Cntd.

Here, we present our results for input poisoning for the Laplace mechanism and the *RRCheck* mechanisms.

Recall in the Laplace mechanism, each user simply reports a private estimate of their degree. Under input poisoning attacks, Laplace noise is added to the poisoned input before it is reported to the data aggregator. Consequently, the response poisoning attack in which a malicious user could *deterministically* report their degree as $n - 1$ (Thm. 6) is no longer possible – in order to manipulate their degree by $n - 1$, the malicious user needs to get lucky with the sampled Laplace noise, resulting in the following theorem:

**Theorem 9.** *Let $\mathcal{M}$ be a set of malicious users with $|\mathcal{M}| = m$. The $R_{Lap}$ protocol has $(\frac{1}{\epsilon} \ln \frac{n}{\delta}, \delta)$ honest error and $(n - 1, \frac{1}{2})$-malicious error with respect to any input poisoning attack from $\mathcal{M}$.*

The proof of the above theorem is in App. G.8. Unsurprisingly, compared to Thm. 6 for response poisoning, the honest error is unchanged because no attack is possible for honest users. However, the malicious error is different. Thm. 6 delineates an $(n - 1)$-tight malicious error, demonstrating the feasibility of the worst-case attack in which a malicious user can *always* manipulate their degree by $n - 1$. In contrast, $R_{Lap}$ has $(n - 1, \frac{1}{2})$-malicious error with respect to any input poisoning attack. This is because the sampled Laplace noise is negative with probability $\frac{1}{2}$. Hence, even a worst-case malicious user who sends the maximum degree of $n - 1$ will only get assigned a final estimate this high if the sampled noise is non-negative. Thus, the noise in the Laplace mechanism prevents the adversary from carrying out the deterministic worst-case attack.

Now we present our results for *RRCheck*:

**Algorithm 4** *Hybrid*: $\{0,1\}^{n \times n} \mapsto \{\mathbb{N} \cup \{\perp\}\}^n$

---

**Parameters:** $\epsilon$ - Privacy parameter;
       $\tau$ - Threshold for consistency check;
**Input:** $\{l_1, \cdots, l_n\}$ where $l_i$ is $U_i$'s adjacency list;
**Output:** $(\hat{d}_1, \cdots, \hat{d}_n)$ where $\hat{d}_i$ is $U_i$'s degree estimate;
**Users**
1: Select $c \in (0,1)$
2: $\rho = \frac{1}{1+e^{c\epsilon}}$
3: **for** $i \in [n]$ **do**
4:   $q_i = RR_\rho(l_i)$
5:   $\tilde{d}_i^{lap} = \|l_i\|_1 + Lap(\frac{1}{(1-c)\epsilon})$
**Data Aggregator**
6: **for** $i \in [n]$ **do**
7:   $count_i^{11} = \sum_{j \in [n] \setminus i} q_i[j] q_j[i]$
8:   $count_i^{01} = \sum_{j \in [n] \setminus i} (1 - q_i[j]) q_j[i]$
9:   **if** $|count_i^{01} - \rho(1-\rho)(n-1)| \leq \tau$ **then**
10:     $\tilde{d}_i^{rr} = \frac{1}{1-2\rho}(count_i^{11} - \rho^2(n-1))$
11:     **if** $|\tilde{d}_i^{rr} - \tilde{d}_i^{lap}| \leq \frac{2\tau}{1-2\rho} + \frac{1}{(1-c)\epsilon} \ln \frac{2n}{\delta}$ **then**
12:       $\hat{d}_i = \tilde{d}_i^{lap}$
13:     **else**
14:       $\hat{d}_i = \perp$
15:   **else**
16:     $\hat{d}_i = \perp$
**return** $(\hat{d}_1, \hat{d}_2, \ldots, \hat{d}_n)$

---

**Theorem 10.** *Let $\mathcal{M}$ be a set of malicious users with $|\mathcal{M}| = m$. The protocol RRCheck run with $\tau = m(1-2\rho) + \sqrt{8 \max\{\rho n, m\} \ln \frac{8n}{\delta}}$ has $(2m + 4\sqrt{\max\{n, m(e^\epsilon + 1)\}} \frac{\sqrt{2(e^\epsilon+1) \ln \frac{8n}{\delta}}}{e^\epsilon - 1}, \delta)$-honest error and malicious error with respect to any input poisoning attack from $\mathcal{M}$.*

The proof is in App. G.10. For typical values of $\epsilon$, the honest error and the malicious error can be written as $(\tilde{O}(m + \frac{\sqrt{n}}{\epsilon}), \delta)$ (because $\sqrt{m(e^\epsilon + 1)} \leq \sqrt{n}$). Compared to Thm. 2 for response poisoning attacks, there is an improvement of $\frac{m}{\epsilon}$ which is a direct consequence of a smaller $\tau$.

# D EVALUATION CNTD.

## D.1 EXPERIMENTAL SETUP

**Datasets.** We consider two graphs – a real-world sparse graph and a synthetically generated dense graph.

- *FB.* This graph corresponds to data from Facebook McAuley & Leskovec (2012) representing the friendships of 4082 Facebook users. The graph has 88K edges.
- *Syn.* To test a more dense regime, we evaluate our protocols on a synthetic graph generated using the Erdos-Renyi model Erdős et al. (1960) with parameters $G(n = 4000, p = 0.5)$ ($n$ is the number of edges; $p$ is the probability of including any edge in the graph). The graph has $\approx 8$ million edges.

**Configurations.** For every attack we report the maximum error over all the honest targets (honest error, $\alpha_1$) and the malicious targets (malicious error, $\alpha_2$). We run each experiment 50 times and report the mean. We use $\delta = 10^{-6}$ and $c = 0.9$ for *Hybrid*.

Our theoretical results suggested setting $\tau = m + C\sqrt{\rho n}$, where $C$ is a constant that is obtained from Chernoff's bounds, for the different input and response manipulation attacks. The constant $C$ is not tight, and for the practical interest of using as small a threshold as possible, we sought to set $\tau$ as small as possible so as not to falsely flag any honest user. Note that lower the threshold, lower is the

| | SimpleRR | | RRCheck | | Hybrid | |
|---|---|---|---|---|---|---|
| | FB | Syn | FB | Syn | FB | Syn |
| Min. | 0 | 0 | 56.0% | 52.0% | 54.5% | 51.0% |
| Mean. | 0 | 0 | 63.2% | 61.4% | 62.1% | 60.0% |
| Max. | 0 | 0 | 75.0% | 70.0% | 70.0% | 70.0% |

Table 1: Table of max, min, and average percentage of malicious targets flagged for degree inflation attacks.

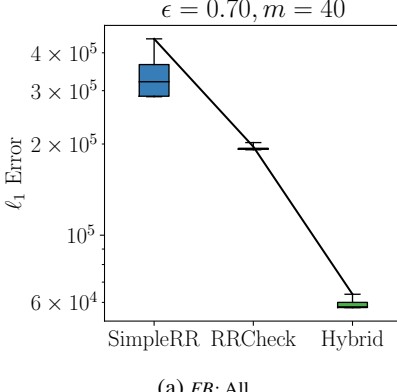

(a) *FB:* All

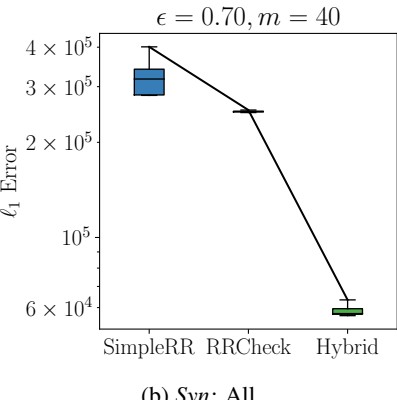

(b) *Syn:* All

Figure 5: Robustness Analysis: The whiskers range from the maximum to the minimum empirical honest error or malicious error observed across all the attacks of the specific type. The line corresponds to the strongest evaluated attack of the specific type.

permissible skew ($\alpha_1$ and $\alpha_2$ for the honest error and the malicious error, respectively) introduced by poisoning, thereby improving the robustness of our protocols. We ran preliminary experiments using 50 runs of each protocol on both graphs, and we found that at all values of $\epsilon$, setting $\tau = m + 0.4\sqrt{\rho n}$ (for $m = 40$) and $\tau = m + 0.1\sqrt{\rho n}$ (for $m = 1500$) did not result in any false positives. Thus, we used these smaller thresholds in our experiments, and throughout the experiments there were no false positives.

**Attacks.** For each attack type, we consider both input and response poisoning versions. In the following, let $U_t$ represent the target user.

*RRCheck.* For the degree inflation attack, the non-target malicious users always report a 1 for the malicious user $U_t$ (i.e. that they are connected to $U_t$) in the hopes of increasing $U_t$'s degree estimate. Likewise, $U_t$ reports 1s for all other malicious users. For the honest users, $U_t$ reports extra 1s (for non-neighbors) in the hopes of further increasing their degree estimate. The exact mechanism depends on whether it is response poisoning or input poisoning and is detailed in App. E.

For degree deflation, we consider the worst-case scenario where $m$ of the neighbors of the honest user $U_t$ act maliciously. The malicious neighbors always report 0 for their edges to $U_t$ (see App. E).

*Hybrid.* For degree inflation, the non-target malicious users report their edges using the same strategy as in *RRCheck*. For $\tilde{d}_i^{lap}$, they send their true degree estimates since their degrees are not the targets. Similarly, $U_t$ uses the same strategy as in *RRCheck* for reporting their edges. For $\tilde{d}_t^{lap}$, $U_t$ reports an inflated value based on the reported edges and the threshold $\tau$ (see App. E). For degree deflation, we consider the worst-case scenario where $m$ of the neighbors of $U_t$ act maliciously. The malicious users behave as they did in *RRCheck* and report their true degrees for $\tilde{d}_i^{lap}$, as these are not the targeted degrees.

*SimpleRR.* For degree inflation, we consider the worst-case scenario where the target malicious user $U_t$ is responsible reporting all their edges, and chooses to reports all 1s. For degree deflation, we again consider the worst case scenario where the malicious users are responsible for reporting the edges to $U_t$, and they report 0s.

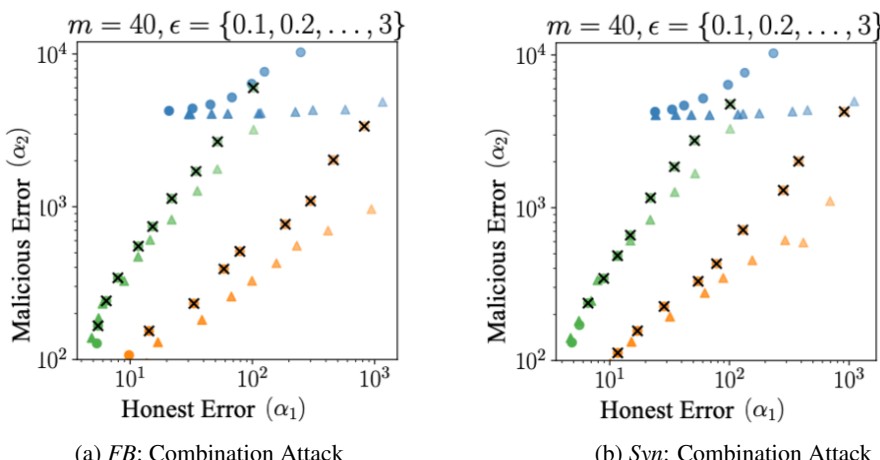

(a) *FB*: Combination Attack    (b) *Syn*: Combination Attack

Figure 6: Robustness analysis with varying $\epsilon$: Higher brightness denotes higher $\epsilon$

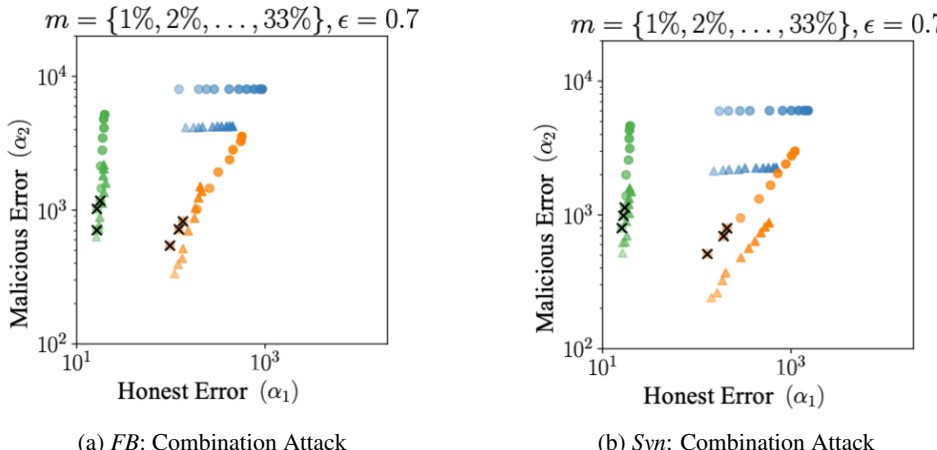

(a) *FB*: Combination Attack    (b) *Syn*: Combination Attack

Figure 7: Robustness analysis with varying number of malicious users $m$: Higher brightness denotes higher $m$

## D.2   EXPERIMENTAL RESULTS CNTD.

$\ell_1$ **Error.** We report the $\ell_1$ error of the entire noisy degree vector $\hat{\mathbf{d}} = \langle \hat{d}_1, \ldots, \hat{d}_n \rangle$ in Fig. 5a and 5b, for *FB* and *Syn*, respectively. We observe that *Hybrid* performs the best. This is because recall that *Hybrid* has the best honest error while its malicious error is comparable to that of *RRCheck*. Since the number of honest users is much higher than the number of malicious users, the $\ell_1$ error of *Hybrid* is significantly better than that of *RRCheck* due to its lower honest error. For instance, for the strongest overall attack we evaluated in terms of the $\ell_1$ error (A8 – the same as that for the degree inflation case) *Hybrid* has $4.0\times$ and $6.3\times$ lower $\ell_1$ error than *RRCheck* and *SimpleRR*, respectively for *Syn*.

## D.3   IMPACT OF ALGORITHMIC PARAMETERS

Here, we study the impact of the three algorithmic parameters – privacy parameter $\epsilon$, total number of malicious users $m$ and threshold for consistency check $\tau$ – on the attack efficacy.

**The effect of $\epsilon$.** Figs. 6b and 6a show the impact of the attacks with varying privacy parameter $\epsilon$. We study the strongest combination attack (attack A10) which considered in the previous section. We observe that, increasing privacy (lower $\epsilon$) leads to more skew for all attacks on all three protocols. For instance, the malicious error of the response poisoning version of the degree deflation attack for *FB* $42\times$ worse for $\epsilon = 0.1$ than that for $\epsilon = 3$ for *Hybrid*. Additionally, we observe that malicious users get flagged only response poisoning since this is a stronger attack than input poisoning.

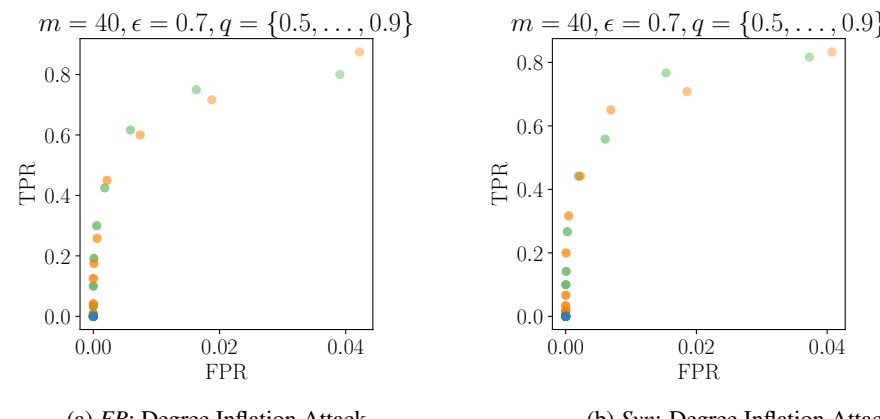

(a) *FB*: Degree Inflation Attack        (b) *Syn*: Degree Inflation Attack

Figure 8: Impact of varying threshold $\tau$ on flagging malicious users: Higher brightness denotes higher $\tau$

**The effect of $m$.** We show how the attack efficacy varies with $m$ in Figs. 7a and 7b. We consider the strongest combination attack (attack A10) here as well. As expected, the impact of poisoning worsens with increasing $m$. Specifically, both the honest error and the malicious error of *RRCheck* worsen with increasing $m$. For instance, the honest error of response poisoning degrades by $8.5\times$ as we increase $m$ from $1\%$ to $33\%$ for *Syn*. The honest error is uninfluenced by $m$ for *Hybrid* – this is because it reports the Laplace estimates for the honest users which are not impacted by the malicious users. On the other hand, the malicious error remains unaffected by $m$ for *SimpleRR* since here a malicious target can always carry-out the worst-case attack regardless of $m$ (Thm. 7). Another interesting observation is that even a relatively small number of malicious parties ($m = 1\%$) can stage significantly damaging poisoning attacks. This demonstrates the practical threat of such poisoning attacks. Nevertheless, our results show that our proposed protocols are able to significantly reduce the impact of poisoning attacks even with a large number of malicious users ($m = 33\%$).

**The effect of $\tau$.** Recall for a degree inflation attack, the target malicious user $U_t$ falsely reports extra 1s for its honest non-neighbors. Now for any given attack, lower the value of the threshold higher is the chance of flagging $U_t$ (true positive rate, TPR). An additional advantage is that this reduces the amount by which $U_t$ can skew its degree estimate (i.e., improves malicious error $\alpha_2$) However, the cost is an increase in the chance of erroneously flagging an honest user (false positive rate, FPR). In order to obtain a rigorous formal guarantee on both true positive rate (honest error) and false positive rate (malicious error), the theoretical thresholds presented in our theorems are quite stringent. However, we observe that in practice we can use lower values of threshold. We show this in Fig. 8a and 8b on the strongest inflation attack evaluated (A11 in Table 2). There is no notion of threshold for the naive protocol *SimpleRR*, hence it has TPR=FPR=0. For our robust protocols, *Hybrid* and *RRCheck*, we test 10 different values of lower thresholds $q \cdot \tau$ where $q \in (0, 1)$. We observe that lowering the threshold delineates a trade-off between TPR and FPR. For all our experiments, we select a threshold that maximizes TPR at FPR=0 through empirical testing.

# E   ATTACKS DETAILS

In this section, we describe the specific implementations of the attacks we use for our evaluation in Section 7.

Recall an attack consists of $m$ malicious users, where $m$ is known beforehand. Each malicious user may perform any of the following three actions: 1) lie about their own connections to changer their estimate, 2) target some subset of malicious users to change those estimates, and 3) target some subset of honest users to tamper with those estimates. We consider sixteen possible ways to do this in ways that would often occur in the real world. The methods appear in Table 2, and we describe them now.

In the simplest attacks (A1 - A3), we consider a set of compromised malicious users whose goal is to either inflate a target malicious user or deflate a target honest user. In A1 and A2, the compromised malicious users constitute a random subset of users. In A3, the compromised malicious users come

from the neighbors of the target honest user, representing a worse attack. In these attacks, and for the rest, we assume the target malicious user lies about his connections to try to increase his degree while the target honest user follows protocol.

In A4 and A5, we scale up A1 to more malicious targets, which could happen if a group of compromised users is used to target multiple accounts. As there is no way to select friends of many targets at once, we omit A3.

In A6 through A8, we consider a more realistic selection strategy: Instead of malicious users drawn completely at random, we consider they are drawn from a *community* in the graph, which is more realistic as similar accounts are often targeted together. We then consider these accounts targeting multiple honest users, and in A8 we simulate what would happen when the malicious users are used recklessly and target a large fraction of the community.

In A9 and A10, we again consider malicious users in a community, but this time they target both malicious and honest users.

Finally, in the rest of the attacks, we consider combinations of two of the previous attacks carried out independently. This is more realistic, as in the real world, different parties of malicious users may collude independently.

| Attack | Number of Malicious Non-targets | Number of Malicious Targets | Number of Honest Targets | Malicious User Selection Strategy | Description |
|---|---|---|---|---|---|
| A1 | 39 | 1 | 0 | Random | Pure degree inflation attack where the malicious users are chosen at random from the entire graph |
| A2 | 40 | 0 | 1 | Random | Pure degree deflation attack where the malicious users are chosen at random from the entire graph |
| A3 | 40 | 0 | 1 | Neighbor | Pure degree deflation attack where the malicious users are neighbors of the honest target |
| A4 | 35 | 5 | 0 | Random | Pure degree inflation attack where the malicious users are chosen at random from the entire graph |
| A5 | 30 | 10 | 0 | Random | Pure degree inflation attack where the malicious users are chosen at random from the entire graph |
| A6 | 40 | 0 | 5 | Community | Pure degree deflation attack where the malicious users and the honest targets are chosen from the same community of the graph |
| A7 | 40 | 0 | 10 | Community | Pure degree deflation attack where the malicious users and honest targets are chosen from the same community of the graph |
| A8 | 40 | 0 | 600 | Community | Pure degree deflation attack where the malicious users and the honest targets are chosen from the same community of the graph |
| A9 | 35 | 5 | 5 | Community | Combination attacks where the malicious users and the honest targets are chosen from the same community of the graph |
| A10 | 30 | 10 | 10 | Community | Combination attacks where the malicious users and the honest targets are chosen from the same community of the graph |
| A11 | (15,15) | (5,5) | (0,0) | (Community,Community) | Degree inflation attack where two sets of 15 non-target malicious users target 5 malicious users independently in two different communities |
| A12 | (10,10) | (10,10) | (0,0) | (Community,Community) | Degree inflation attack where where two sets of 10 non-target malicious users target 10 malicious users independently in two different communities |
| A13 | (20,20) | (0,0) | (5,5) | (Community,Community) | Degree deflation attack where two sets of 20 malicious users target 5 honest users independently in two different communities |
| A14 | (20,20) | (0,0) | (10,10) | (Community,Community) | Degree deflation attack where two sets of 20 malicious users target 10 honest targets independently in two different communities |
| A15 | (15,20) | (5,0) | (0,5) | (Community,Community) | Combination attacks where two sets of malicious users of sizes 15 and 20 target 10 malicious and 10 honest targets independently in two different communities |
| A16 | (10,20) | (10,0) | (0,10) | (Community,Community) | Combination attacks where two sets of malicious users of sizes 10 and 20 target 10 malicious and 10 honest targets independently in two different communities |

Table 2: Summary of evaluated attacks

### E.1 ATTACKS AGAINST *RRCheck*

#### E.1.1 DEGREE INFLATION ATTACKS

Let $U_t, t \in \mathcal{M}$ denote the target malicious user.
**Input Poisoning.** In this attack, the non-target malicious users set the bit for $U_t$ to be 1. The target malicious user constructs his input by setting 1 for all other malicious users. They also report 1 for honest users to which they share an edge.

For honest users to which $U_t$ does not share an edge, $U_t$ flips some of the bits to 1 with the hopes of artificially increasing his degree. He does this for a $r_1$-fraction of these neighbors. See Algorithm 5 for the details; we term this attack $A_{RRCheck}^{inp}$. Note that if $r_1 = 0$, then the malicious user is being completely honest for these users and will not inflate his degree, and if $r_1 = 1$, then he lies about each of these users and will likely be caught. Thus, his strategy is to pick a value in between 0 and 1, and in the experiments we found that $r_1 = 15\%$ was a good tradeoff point.

---

**Algorithm 5** $A_{RRCheck}^{inp} : \{0,1\}^n \mapsto \{0,1\}^n$

---

    **Parameters:** $\epsilon$ - Privacy parameter;
    **Input:** $l \in \{0,1\}^n$ - True adjacency list;
          $t$ - Target honest user;
    **Output:** $q \in \{0,1\}^n$ - Reported adjacency list;

1:  Select $r_1 \in [0,1]$
2:  $\mathcal{H}_1 = \{i \in \mathcal{H} | l[i] = 1\}$
                                       $\triangleright$ $\mathcal{H}_1$ is the set of honest users with a mutual edge
3:  $\mathcal{H}_0 = \mathcal{H} \setminus \mathcal{H}_1$
                                 $\triangleright$ $\mathcal{H}_0$ is the set of honest users without a mutual edge
4:  $F \in_R \mathcal{H}_0, |F| = r_1|\mathcal{H}_0|$
                                    $\triangleright$ Randomly sample $r_1$ fraction of the users in $\mathcal{H}_0$
5:  $l' = \{0, 0, \cdots, 0\}$
6:  **for** $i \in \mathcal{H}_1 \cup \mathcal{M} \cup F$ **do**
7:     $l'[i] = 1$
8:  **for** $i \in [n]$ **do**
9:     $q[i] = RR_\rho(l'[i])$
    **return** $q$

---

 

---

**Algorithm 6** $A_{RRCheck}^{resp} : \{0,1\}^n \mapsto \{0,1\}^n$

---

    **Parameters:** $\epsilon$ - Privacy parameter;
    **Input:** $l \in \{0,1\}^n$ - True adjacency list;
    **Output:** $q \in \{0,1\}^n$ - Reported adjacency list;

1:  Select $r_1 \in [0,1]$
2:  $q = \mathsf{RR}_\rho(l)$
3:  $\mathcal{I}_1 = \{i \in \mathcal{H} | q[i] = 1\}$
                                     $\triangleright$ $\mathcal{H}_1$ is the set of honest users with an edge in $q$
4:  $\mathcal{I}_0 = \mathcal{H} \setminus \mathcal{I}_1$
5:  $F \in_R \mathcal{I}_0, |F| = r_1|\mathcal{I}_0|$
                                    $\triangleright$ Randomly sample $r_1$ fraction of the users in $\mathcal{I}_0$
6:  **for** $i \in \mathcal{I}_1 \cup \mathcal{M} \cup F$ **do**
7:     $q[i] = 1$
    **return** $q$

---

**Response Poisoning.** For response poisoning, the non-target malicious first find a plausible response by applying $RR_\rho$ to their data. They then set the bit for $U_t$ to be 1, indicating they are connected to this user.

The target malicious user constructs his response by first applying $RR_\rho$ to his data to compute a plausible response. Then, he flips his bits to malicious users to 1, and for honest users, he takes a $r_1$-fraction of the 0s in his response and flips them to 1. The quantity $r_1$ is a tradeoff parameter with the same intuition as for $A_{RRCheck}^{inp}$. The details of this attack appear in Algorithm 6, and it is termed $A_{RRCheck}^{resp}$.

### E.1.2   Degree Deflation Attacks

Let $U_t, t \in \mathcal{H}$ denote the target honest user.
**Input Poisoning.** Here, every malicious user constructs his input acting honestly for non-target users and setting a 0 for $U_t$.
**Response Poisoning.** Every malicious user acts honestly for non-target users by applying randomized response to their input. They finally send a 0 for their connection to $U_t$.

---

**Algorithm 7** $A_{Hybrid}^{inp} : \{0,1\}^n \mapsto \{0,1\}^n$

---

    **Parameters:** $\epsilon$ - Privacy parameter;
    **Input:** $l \in \{0,1\}^n$ - True adjacency list;
    **Output:** $q \in \{0,1\}^n$ - Reported adjacency list;
        $\tilde{d}^{lap}$ - Reported noisy degree estimate;

1: Select $r_2 \in [0,1]$
2: $\rho = \frac{1}{1+e^{c\epsilon}}$
        $\triangleright$ $c$ is the constant used in Alg. 4 to divide the budget between the RR and Laplace steps.
3: $q \leftarrow A_{RRCheck}^{inp}(l, c\epsilon)$
4: $\tilde{count}^{11} \leftarrow m(1-\rho)^2 + \mathbb{E}[\sum_{i \in \mathcal{H}} q_i RR_\rho(l_i)]$
5: $e^{rr,inp} = \frac{\tilde{count}^{11} - \rho^2 n}{1-2\rho}$.
6: $\hat{d}^{Lap} = e^{rr,inp} + r_2 \frac{\tau}{1-2\rho} + \eta$ where $\eta \sim Lap(\frac{1}{(1-c)\epsilon})$.
    **return** $q, \tilde{d}^{Lap}$

---

### E.2 ATTACKS AGAINST *Hybrid*

#### E.2.1 DEGREE INFLATION ATTACKS

Let $U_t, t \in \mathcal{M}$ be the target malicious user.
**Input Poisoning.** The non-target malicious users flip their edge to $U_t$ to a 1 as they do in $A_{RRCheck}^{inp}$. They send an honest estimate of their degree $\tilde{d}^{Lap}$ as this does not affect the target.

The target malicious user crafts his input adjacency list $q$ as he did in $A_{RRCheck}^{inp}$. For his estimate $\tilde{d}_t^{Lap}$, he computes the expected value of $\tilde{d}_t^{rr}$ given that he submitted $q$ while the other users either submit $RR_\rho(l_i)$ or $RR_\rho(1)$, depending if they are honest or malicious. Specifically, the expected value is given by

$$e^{rr,inp} = \frac{m(1-\rho)^2 + \mathbb{E}[\sum_{i \in \mathcal{H}} q_i RR_\rho(l_i)] - \rho^2 n}{1-2\rho}.$$

He finally sets $\tilde{d}_t^{rr} = e_t^{rr,inp} + r_2 \frac{\tau}{1-2\rho}$ where $r_2 \in [0,1]$, which again trades off between how much cheating is possible and getting flagged. During the trials, we used $q_2 = 0.1$ as this did not significantly increase the target's chance of being rejected as $\perp$. This attack, termed $A_{Hybrid}^{inp}$, appears in Algorithm 7.

**Response Poisoning.** The non-target malicious users flip their edge to $U_t$ to a 1 as they do in $A_{RRCheck}^{inp}$. They send an honest estimate of their degree $\tilde{d}^{Lap}$ as this does not affect the target.

The target malicious user crafts his response adjacency list $q$ as he did in $A_{RRCheck}^{resp}$. For his estimate $\tilde{d}_t^{Lap}$, he computes the expected value of $\tilde{d}_t^{rr}$ given that he submitted $q$ while the other users either submit $RR_\rho(l_i)$ or 1, depending if they are honest or malicious. This expected value is given by

$$e^{rr,resp} = \frac{m + \mathbb{E}[\sum_{i \in \mathcal{H}} q_i RR_\rho(l_i) - \rho^2 n]}{1-2\rho}.$$

He finally sets $\tilde{d}_t^{rr} = e^{rr,resp} + r_2 \frac{\tau}{1-2\rho}$ where $r_2 \in [0,1]$ serves a similar tradeoff purpose as for $A_{Hybrid}^{inp}$.

#### E.2.2 DEGREE DEFLATION ATTACKS

Let $U_t, t \in \mathcal{H}$ represent the honest target.
**Input Poisoning.** For the adjacency list, all the malicious users follow the same protocol as for *RRCheck*. For the degree, all the malicious users follow the Laplace mechanism truthfully as these values are immaterial for estimating the degree of the target honest user.

**Response Poisoning.** For the adjacency list, all the malicious users follow the same protocol as for *RRCheck*($\cdot$). For the degree, all the malicious users follow the Laplace mechanism truthfully as these values are immaterial for estimating the degree of the target honest user.

---

**Algorithm 8** $A_{Hybrid}^{resp} : \{0,1\}^n \mapsto \{0,1\}^n$

---

**Parameters:** $\epsilon$ - Privacy parameter
**Input:** $l \in \{0,1\}^n$ - True adjacency list;
**Output:** $q \in \{0,1\}^n$ - Reported adjacency list;
$\quad\quad \tilde{d}^{lap}$ - Reported noisy degree estimate;

1: Select $r_2 \in [0,1]$
2: $q = A_{RRCheck}^{resp}(l, \epsilon)$
3: $\rho = \frac{1}{1+e^{c\epsilon}}$
$\quad \triangleright c$ determines how the privacy budget is divided between the two types of response as in Alg. 4
4: $\tilde{count}^{11} \leftarrow m + \mathbb{E}[\sum_{i \in \mathcal{H}} q_i RR_\rho(l_i)]$
5: $e^{rr,resp} = \frac{m+\tilde{count}^{11}-\rho^2 n}{1-2\rho}$
6: $\tilde{d}^{Lap} = e^{rr,resp} + r_2 \frac{\tau}{1-2\rho}$
**return** $q, \tilde{d}^{Lap}$

---

# F    RELATED WORK CNTD.

A recent line of work Cheu et al. (2021); Cao et al. (2021); Wu et al. (2021); Li et al. (2022) has explored the impact of poisoning in LDP. However, these works focused either on tabular data or key-value data. Additionally, prior work mostly focuses on the task of frequency estimation which is different from our problem of degree estimation. For the former, each user has some item from an input domain and the data aggregator wants to compute the histogram over all the users' items. Whereas, we compute the degree vector $\langle \hat{d}_1, \dots, \hat{d}_n \rangle$ – each user directly reports their degree $d_i$ (a count or via an adjacency list). More specifically, Cao et al. Cao et al. (2021) proposed attacks where an adversary could increase the estimated frequencies for adversary-chosen target items or promote them to be identified as heavy hitters. Wu et al. Wu et al. (2021) extended the attacks for key-value data where an adversary aims to simultaneously promote the estimated frequencies and mean values for some adversary-chosen target keys. Cheu et al. Cheu et al. (2021) formally analyzed the poisoning attacks on categorical data and showed that local algorithms are highly vulnerable to adversarial manipulation – when the privacy level is high or the input domain is large, an adversary who controls a small fraction of the users in the protocol can completely obscure the distribution of the users' inputs. This is essentially an impossibility result for robust estimation of categorical data via non-interactive LDP protocols. Additionally, they showed that poisoning the noisy messages can be far more damaging than poisoning the data itself. A recent work Li et al. (2022) studies the impact of data poisoning for mean and variance estimation for tabular data. In the shuffle DP model, Cheu et al. Cheu & Zhilyaev (2022) have studied the impact of poisoning on histogram estimation. In terms of general purpose defenses, prior work has explored strategies strategy based on cryptographically verifying implementations of LDP randomizers Kato et al. (2021); Ambainis et al. (2003); Moran & Naor (2006) – this would restrict the attacks to input poisoning only.

# G    PROOFS

First, we introduce notation and preliminary results used in our proofs.

## G.1    NOTATION

In this section, for a graph $G$ with vertices $[n]$, we let $d_i(S)$ for $S \subseteq [n]$ denote the number of neighbors of node $i$ in the set $S$. We will often abuse notation for a set $\mathcal{S}$ of users by also letting $\mathcal{S}$ be the indices of the users in the set. Thus, we may let $i \in \mathcal{S}$ be the index of some user in $\mathcal{S}$. Finally, we sometimes refer to user $U_i$ simply as user $i$.

## G.2    PRELIMINARY RESULTS

We will heavily make use of the following concentration result:

**Lemma 1.** *Let $X_1, \ldots, X_n$ denote independent random variables such that $X_i \sim \text{Bernoulli}(p_i)$. Let $v = \sum_{i=1}^{n} p_i(1 - p_i)$, and $X = \sum_{i=1}^{n} X_i$. Then,*

$$\Pr[|X - \mathbb{E}[X]| \geq \max\{1.5 \ln \frac{2}{\delta}, \sqrt{2v \ln \frac{2}{\delta}}\}] \leq \delta.$$

*Proof.* Center the random variables so that $Z_i = X_i - p_i$; the variance $v$ does not change. We know by Bernstein's inequality that for all $t \geq 0$,

$$\Pr[Z \geq t] \leq \exp\left(\frac{-t^2}{2(v + t/3)}\right) \leq \exp\left(-\max\left\{\frac{t^2}{2v}, \frac{3t}{2}\right\}\right).$$

Thus, if $t \geq \max\{\frac{3}{2} \ln \frac{2}{\delta}, \sqrt{2v \ln \frac{2}{\delta}}\}$, then $\Pr[Z \geq t] \leq \frac{\delta}{2}$. Applying the argument to $-Z$, we obtain the two-sized bound.

$\square$

Next, we observe the following facts about randomized response.

**Fact 1.** *If user $i \in \mathcal{H}$, then $\mathbb{E}[q_i[j]] = \rho + (1 - 2\rho)d_i(j)$.*

**Fact 2.** *If users $i, j \in \mathcal{H}$, then $\mathbb{E}[q_i[j]q_j[i]] = \rho^2 + (1 - 2\rho)d_i(j)$.*

### G.3 PROOF OF THEOREM 6

Recall that in the Laplace mechanism, a user's degree estimate $\hat{d}_i$ is simply $d_i + L_i$, where $L_i \sim Lap(\frac{1}{\epsilon})$ is a Laplace random variable generated by the user.

**Honest Error.** The honest error guarantee follows from the concentration of Laplace distribution: Each Laplace random variable $L_i$ satisfies $|\Pr[|L_i| \geq t] \leq e^{-t\epsilon}$. Setting $t = \frac{1}{\epsilon} \ln \frac{n}{\delta}$ and applying the union bound, each of the $n$ Laplace variables will satisfy $|L_i| \leq \frac{1}{\epsilon} \ln \frac{\delta}{n}$ with probability $1 - \delta$, and if this holds, then $|d_i - \hat{d}_i| \leq \frac{1}{\epsilon} \ln \frac{\delta}{n}$ for honest users.

**Tight Malicious Error.** Consider the empty graph. A malicious user $U_i$ may report $n - 1$, the maximum possible degree, and thus $\hat{d}_i = n - 1$ while $d_i = 0$.

### G.4 PROOF OF THEOREM 7

**Honest Error.**

As defined in *SimpleRR*, the estimator $count_i^1$ is given by

$$count_i^1 = \left(\sum_{j < i} q_j[i] + \sum_{i < j} q_i[j]\right) \tag{5}$$

We may alternatively split the above sum into honest bits and malicious bits as $count_i^1 = hon_i + mal_i$. Here,

$$hon_i = \sum_{j < i, j \in \mathcal{H}} q_j[i] + \sum_{i < j} q_i[j]$$

$$mal_i = \sum_{j < i, j \in \mathcal{M}} q_j[i].$$

Since all bits in the sum $hon_i$ are honest, by Fact 1 we have $\mathbb{E}[hon_i] = \rho|\mathcal{H}_i| + (1 - 2\rho)d_i(\mathcal{H}_i)$, where $\mathcal{H}_i = \mathcal{H} \cup \{1, 2, \ldots, i - 1\}$.

Furthermore, $0 \leq mal_i \leq |\mathcal{M}_i|$, where $\mathcal{M}_i = [n] \setminus \mathcal{H}_i$. This implies $|mal_i - E_{mal,i}| \leq |\mathcal{M}_i|$, where $E_{mal,i} = \rho|\mathcal{M}_i| + (1 - 2\rho)d_i(\mathcal{M}_i)$. By Lemma 1 and a union bound, with probability $1 - \delta$,

we have for all $i \in \mathcal{H}_i$ that

$$|hon_i - \mathbb{E}[hon_i] + mal_i - E_{mal,i}| \leq \sqrt{2\rho n \ln \frac{2n}{\delta}} + |\mathcal{M}_i|$$

$$\implies |count_i^1 - \rho n - (1 - 2\rho)d_i| \leq \sqrt{2\rho n \ln \frac{2n}{\delta}} + m$$

$$\implies |\hat{d}_i - d_i| \leq \frac{1}{1 - 2\rho}\sqrt{2\rho n \ln \frac{2n}{\delta}} + \frac{m}{1 - 2\rho}.$$

**Tight Malicious Error.** Consider the empty graph, and suppose that user $n$ is malicious. Since this user reports all his edges, he may report $q_i[j] = 1$ for all $j < 1$. Thus, $\hat{d}_n \geq n - 1$, but $d_n = 0$, showing $n - 1$-tight malicious error.

G.5  PROOF OF THEOREM 2

Recall the key quantities defined in *RRCheck* (Algorithm 1):

$$count_i^{11} = \sum_{j \in [n] \setminus i} q_i[j]q_j[i] \tag{6}$$

$$count_i^{01} = \sum_{j \in [n] \setminus i} (1 - q_i[j])q_j[i]. \tag{7}$$

We now prove honest error.

**Honest Error.** It will be helpful to split $count_i^{11} = hon_i^{11} + mal_i^{11}$, where $hon_i^{11} = \sum_{j \in \mathcal{H} \setminus i} q_i[j]q_j[i]$ and $mal_i^{11} = \sum_{j \in \mathcal{M} \setminus i} q_{ij}q_{ji}$. We define $hon_i^{01}$ and $mal_i^{01}$ similarly such that they satisfy $count_i^{01} = hon_i^{01} + mal_i^{01}$. We break the proof into two claims: showing that honest users receive an accurate estimate and that they are not disqualified.

**Claim 1.** *We have*
$$\Pr[\forall U_i \in \mathcal{H}. \ |\hat{d}_i - d_i| \geq \tfrac{m + 2\sqrt{\rho n \ln \frac{4n}{\delta}}}{1 - 2\rho}] \leq \frac{\delta}{2}.$$

*Proof.* Let $U_i \in \mathcal{H}$. Then, $hon_i^{11}$ is a sum of $h - 1$ Bernoulli random variables with $p = \rho^2$ or $(1 - \rho)^2$. By Fact 2, we have

$$\mathbb{E}[hon_i^{11}] = \rho^2(h - 1) + (1 - 2\rho)d_i(\mathcal{H})$$

Now, $v$ defined in Lemma 1 satisfies $(h - 1)\rho^2 \leq v \leq (h - 1)(1 - (1 - \rho)^2) \leq 2(h - 1)\rho$. Applying the Lemma and a union bound, we have with probability at least $1 - \frac{\delta}{2}$ that for all $i \in \mathcal{H}$,

$$|hon_i^{11} - \mathbb{E}[hon_i^{11}]| \leq 2\sqrt{(h - 1)\rho \ln \frac{4n}{\delta}}. \tag{8}$$

On the other hand, we have that $0 \leq mal_i^{11} \leq m$, so if we let $E_{mal,i}^{11} = \rho^2 m + (1 - 2\rho)d_i(\mathcal{M})$ (defined for convenience later), then $|mal_i^{11} - E_{mal,i}^{11}| \leq m$.

Applying the triangle inequality, the following holds over all $i \in \mathcal{H}$:

$$|hon_i^{11} - \mathbb{E}[hon_i^{11}] + mal_i^{11} - E_{mal,i}^{11}| \leq m + 2\sqrt{\rho n \ln \frac{4n}{\delta}}$$

$$\implies |count_i^{11} - \rho^2(n - 1) - (1 - 2\rho)d_i| \leq m + 2\sqrt{\rho n \ln \frac{4n}{\delta}}$$

$$\implies |\hat{d}_i - d_i| \leq \frac{m + 2\sqrt{\rho n \ln \frac{4n}{\delta}}}{1 - 2\rho}$$

This proves the claim. □

Next, we show that honest users are not likely to be disqualified.

**Claim 2.** *We have*

$$\Pr[\forall U_i \in \mathcal{H}. \ |count_i^{01} - \rho(1-\rho)(n-1)| \geq \tau] \leq \frac{\delta}{2},$$

where $\tau = m + \sqrt{2\rho n \ln \frac{4n}{\delta}}$

*Proof.* Let $U_i$ be honest. Then, the quantity $hon_i^{01}$ consists of $h-1$ Bernoulli random variables drawn from $\rho(1-\rho)$. We have

$$\mathbb{E}[hon_i^{01}] = \rho(1-\rho)(h-1).$$

As defined in Lemma 1, $v$ satisfies $\frac{1}{2}(h-1)\rho \leq P \leq (h-1)\rho$. Applying the Lemma and a union bound, we have with probability $1 - \frac{\delta}{2}$ that for all $i \in \mathcal{H}$,

$$|hon_i^{01} - \mathbb{E}[hon_i^{01}]| \leq \sqrt{2\rho(h-1) \ln \frac{4n}{\delta}} \tag{9}$$

Noticing that $|mal_i^{01} - m\rho(1-\rho)| \leq m$, we have by the triangle inequality that

$$|count_i^{01} - \rho(1-\rho)(n-1)| \geq m + \sqrt{2\rho n \ln \frac{4n}{\delta}}.$$

This concludes the proof. $\qquad\square$

Putting it together, $\left( m(\frac{e^\epsilon+1}{e^\epsilon-1}) + \sqrt{n}\frac{2\sqrt{(e^\epsilon+1)\ln\frac{4n}{\delta}}}{e^\epsilon-1}, \delta \right)$-honest error follows.

**Malicious Error.**

When player $i$ is a malicious player, we can still prove a tight bound on $count_i^{11} + count_i^{01}$, and this combined with the check in *SimpleRR* means that his degree estimate will be accurate.

**Claim 3.** *We have*

$$\Pr[\forall i \in \mathcal{M}. \ |count_i^{11} + count_i^{01}$$
$$- (1-2\rho)d_i - \rho(n-1)| \leq \tau] \geq 1 - \delta,$$

*where* $\tau = m + \sqrt{2\rho n \ln \frac{4n}{\delta}}$.

*Proof.* Observe that $count_i^{11} + count_i^{01} = \sum_{j=1, j\neq i}^{n} q_j[i]$. Let $hon_i^1$ denote the sum of the $q_j[i]$ where $j$ is honest, and $mal_i^1$ denote the sum of the malicious players. By Fact 1, we have $\mathbb{E}[hon_i^1] = d_i(\mathcal{H})(1-2\rho) + h\rho$. Applying a union bound over Lemma 1, for all $i \in \mathcal{M}$, we have with probability at most $\delta$ that

$$|hon_i^1 - \mathbb{E}[hon_i^1]| \geq \sqrt{2\rho n \ln \frac{2m}{\delta}} \tag{10}$$

Because $|mal_i^1 - (1-2\rho)d_i(\mathcal{M}) - \rho(m-1)| \leq m$, the claim follows from the triangle inequality. $\quad\square$

To conclude the proof, consider any malicious user $i \in \mathcal{M}$ is not disqualified ($\hat{d}_i \neq \perp$), as if he is then the malicious error event trivially happens. Thus, it must be true that $|count_i^{01} - (n-1)\rho(1-\rho)| \leq \tau$. However, given this and the event in Claim 3 holds, it follows by the triangle inequality that

$$|count_i^{11} - (1-2\rho)d_i - \rho^2(n-1)| \leq 2\tau$$
$$|\hat{d}_i - d_i| \leq \frac{2\tau}{1-2\rho}$$

This establishes $\left( 2m(\frac{e^\epsilon+1}{e^\epsilon-1}) + 4\sqrt{n}\frac{\sqrt{(e^\epsilon+1)\ln\frac{4n}{\delta}}}{e^\epsilon-1}, \delta \right)$-malicious error.

## G.6 Proof of Theorem 8

**Honest Error.** Let honest $U_i$ share an edge with all malicious users in $\mathcal{M}$. Now, all the malicious users can lie and report 0 for $U_i$, i.e., set $q_j[i] = 0 \forall j \in \mathcal{M}$. This deflates $U_i$'s degree by $m$.

**Malicious Error.** Let malicious $U_i$ share an edge with all users in the graph. Consider the attack where $U_i$ and $U_j, j \in \mathcal{M} \setminus i$ report 0 for their edges, and additionally $U_i$ reports 0 for $\min\{m - 1, n - m\}$ additional honest users. In this way, $U_i$ can deflate its degree estimate by $\min(2m - 1, n)$.

## G.7 Proof of Theorem 3

**Honest Error.** By Claim 2, the first check in *Hybrid* will not set $\hat{d}_i = \perp$ for any honest user with probability at least $1 - \frac{\delta}{4}$. The variables $\hat{d}_i^{rr}$ in *Hybrid* behave identically to $\hat{d}_i$ in *RRCheck*. By Claim 1 we have for all users, $|\hat{d}_i^{rr} - d_i| \leq \frac{m + 2\sqrt{\rho n \ln \frac{8n}{\delta}}}{1 - 2\rho}$, with probability at least $1 - \frac{\delta}{4}$.

By concentration of Laplace random variables, we have for all $i \in \mathcal{H}$ that $|\hat{d}_i^{lap} - d_i| \leq \frac{1}{\epsilon} \ln \frac{2n}{\delta}$ with probability at least $1 - \frac{\delta}{2}$, and by the triangle inequality we have $|\hat{d}_i^{lap} - \hat{d}_i^{rr}| \leq \frac{m + 2\sqrt{\rho n \ln \frac{8n}{\delta}}}{1 - 2\rho} + \frac{1}{\epsilon} \ln \frac{2n}{\delta}$. Thus, the second check will not set $\hat{d}_i = \perp$ assuming these events hold, and the estimator $\hat{d}_i$ satisfies the honest error bound of $\hat{d}_i^{lap}$.

**Malicious Error.** Following the same argument we saw in the proof for malicious error for Theorem 2, we can have that, with probability at least $1 - \frac{\delta}{2}$, for all malicious users $i \in \mathcal{M}$, we have $|\tilde{d}_i^{rr} - d_i| \leq \frac{2\tau}{1 - 2\rho}$. Suppose that $\hat{d}_i$ is not set to be $\perp$. This implies that $|\tilde{d}_i^{rr} - \hat{d}_i^{lap}| \leq \frac{2\tau}{1 - 2\rho} + \frac{1}{\epsilon} \log \frac{2n}{\delta}$. By the triangle inequality, this implies

$$|\tilde{d}_i^{rr} - d_i| \leq \frac{4\tau}{1 - 2\rho} + \frac{1}{\epsilon} \log \frac{2n}{\delta}.$$

This establishes $(\frac{4\tau}{1 - 2\rho} + \frac{1}{\epsilon} \log \frac{2n}{\delta}, \delta)$-malicious error.

## G.8 Proof of Theorem 9

**Honest Error.** The honest error guarantee follows in the same way as Theorem 6.

**Malicious Error** Consider a malicious user $U_i$, and let $m_i$ be the malicious degree estimate sent by $U_i$, with $0 \leq m_i \leq n - 1$. The estimator is given by $\hat{d}_i = m_i + \eta, \eta \sim Lap(\frac{1}{\epsilon})$. Thus, $\Pr[|d_i - m_i - \eta| \geq n - 1] \leq \Pr[\eta > 0] \leq \frac{1}{2}$.

## G.9 Proof of Theorem 4

**Honest Error.** We follow the honest error proof of Theorem 7, with the following change. Observe that $mal_i$ consists of $|\mathcal{M}_i|$ Bernoulli random variables of mean either $\rho$ or $1 - \rho$. Thus, with probability $1 - \frac{\delta}{2}$, we have $|mal_i - \mathbb{E}[mal_i]| \leq \sqrt{2m \ln \frac{4m}{\delta}}$ for all $i \in \mathcal{M}$.

Thus, we can show $|mal_i - E_{mal,i}| \leq (1 - 2\rho)|\mathcal{M}_i|$, where $E_{mal,i} = \rho|\mathcal{M}_i| + (1 - 2\rho)d_i(M_i)$. Finishing the proof, we can show

$$|\hat{d}_i - d_i| \leq \frac{1}{1 - 2\rho}(\sqrt{2\rho n \ln \frac{4n}{\delta}} + \sqrt{2m \ln \frac{4m}{\delta}}) + m.$$

**Malicious Error.**

In order for $|d_i - \hat{d}_i| = n - 1$, it is necessary for $|count_i^1 - \rho(n - 1) - (1 - 2\rho)d_i| \geq (1 - 2\rho)(n - 1)$. We have $count_i^1$ is a sum of $n - 1$ Bernoulli random variables of mean either $\rho$ or $1 - \rho$, so it can be written as $\mu + Z_i$, where $Z_i$ is approximately a normal random variable of mean 0. Observe that, since $\mu$ and $\rho(n - 1) + (1 - 2\rho)d_i$ are in the interval $[\rho(n - 1), (1 - \rho)(n - 1)]$, it is impossible for the difference $\mu - \rho(n - 1) + (1 - 2\rho)d_i$ to exceed $(1 - 2\rho)(n - 1)$ unless $Z_i$ has the correct sign, which happens with probability at most $\frac{1}{2}$. This establishes $(n - 1, \frac{1}{2})$-malicious error.

## G.10 PROOF OF THEOREM 10

**Honest Error.** Our proof follows that of Theorem 2. We are able to prove stronger versions of the claims.

**Claim 4.** *We have*

$$\Pr[\forall i \in \mathcal{H}.\ |\hat{d}_i - d_i| \geq m + \frac{\sqrt{8 \max\{\rho n, m\} \ln \frac{8n}{\delta}}}{1 - 2\rho}] \leq \frac{\delta}{2}.$$

*Proof.* We can control $hon_i^{11}$ in exactly the same way as in Claim 1, so (8) holds with probability $1 - \frac{\delta}{4}$, for all $i \in \mathcal{H}$. On the other hand, we know that $mal_i^{11}$ is now a sum of $d_i(\mathcal{M})$ Bernoulli random variables with bias either $(1 - \rho)^2$ or $(1 - \rho)\rho$, plus a sum of $m - d_i(\mathcal{M})$ Bernoulli random variables with bias either $\rho(1 - \rho)$ or $\rho^2$. Thus,

$$\rho(1 - 2\rho)d_i(\mathcal{M}) + \rho^2 m \leq \mathbb{E}[mal_i^{11}]$$
$$\leq (1 - \rho)(1 - 2\rho)d_i(\mathcal{M}) + \rho(1 - \rho)m.$$

From this, we can show $|\mathbb{E}[mal_i^{11}] - E_{mal,i}^{11}| \leq (1 - 2\rho)m$, where $E_{mal,i}^{11} = \rho^2 m + (1 - 2\rho)d_i(\mathcal{M})$. Applying Hoeffding's inequality, we conclude that with probability at least $1 - \frac{\delta}{4}$, for all $i \in \mathcal{H}$,

$$|mal_i^{11} - \mathbb{E}[mal_i^{11}]| \geq \sqrt{2m \ln \frac{8n}{\delta}}$$

Thus, $|mal_i^{11} - E_{mal,i}^{11}| \leq (1 - 2\rho)m + \sqrt{2m \ln \frac{8n}{\delta}}$. Applying the triangle inequality, we obtain

$$\Pr[|hon_i^{11} + mal_i^{11} - \mathbb{E}[hon_i^{11}] - E_{mal,i}^{11}|$$
$$\geq \sqrt{2m \ln \frac{8n}{\delta}} + (1 - 2\rho)m + 2\sqrt{\rho n \ln \frac{8n}{\delta}}] \leq \frac{\delta}{2}.$$

The result follows in the same way as in Claim 1. $\qquad\square$

**Claim 5.** *We have*

$$\Pr[\forall i \in \mathcal{H}.\ |count_i^{01} - \rho(1 - \rho)(n - 1)| \geq \tau] \leq \frac{\delta}{2},$$

*where $\tau = m(1 - 2\rho) + \sqrt{8 \max\{\rho n, m\} \ln \frac{8n}{\delta}}$.*

*Proof.* We can follow the same line of reasoning as Claim 2 and conclude that (9) holds. Similar to Claim 4, we can show that $|mal_i^{01} - \rho(1 - \rho)m| \leq (1 - 2\rho)m + \sqrt{2m \ln \frac{8n}{\delta}}$ with probability at least $\frac{\delta}{4}$, and applying the triangle inequality, we see

$$\Pr[|count_i^{01} - \rho(1 - \rho)n| \geq m(1 - 2\rho) +$$
$$\sqrt{2m \ln \frac{8n}{\delta}} + \sqrt{2\rho n \ln \frac{8n}{\delta}}] \leq \frac{\delta}{2}.$$
$$\square$$

The $(2m + \frac{4\sqrt{2 \max\{\rho n, m\} \ln \frac{8n}{\delta}}}{1 - 2\rho}, \delta)$-honest error guarantee follows from the union bound over the two claims.

**Malicious Error** When player $i$ is a malicious player, he is still subject to the following claim:

**Claim 6.** *We have*

$$\Pr[\forall i \in \mathcal{M}.\ |count_i^{11} + count_i^{01}$$
$$- (1 - 2\rho)d_i - \rho(n - 1)| \leq \tau|] \geq 1 - \delta,$$

*where $\tau = m(1 - 2\rho) + \sqrt{8 \max\{\rho n, m\} \ln \frac{8n}{\delta}}$.*

*Proof.* Observe that $count_i^{11} + count_i^{01} = \sum_{j=1, j \neq i}^{n} q_j[i] = hon_i^1 + mal_i^1$. With the same argument as in Claim 3, we know that (10) holds. Similarly, each random variable in $mal_i^1$ comes from either Bernoulli($\rho$) or Bernoulli($1 - \rho$), and thus with probability at least $1 - \frac{\delta}{2}$, for all $i \in \mathcal{M}$

$$|mal_i^1 - \mathbb{E}[mal_i^1]| \leq \sqrt{2m \ln \frac{4m}{\delta}}$$

Since $\mathbb{E}[mal_i^1] \in [\rho m, (1 - \rho)m]$, This implies that $|mal_i^1 - \rho m| \leq (1 - 2\rho)m + \sqrt{2m \ln \frac{4m}{\delta}}$. Thus, the claim follows. $\qquad\square$

Having established this claim, we can prove $(2m + 4\sqrt{2 \max\{\rho n, m\} \ln \frac{8n}{\delta}}, \delta)$-malicious error using an identical method as in the proof of malicious error for Theorem 2.

## G.11 PROOF OF THEOREM 5

**Honest Error.** As input manipulation attacks are a subset of response manipulation attacks, the same honest error guarantee as Theorem 3 holds.

**Malicious Error.** The proof of this is similar to the proof of malicious error for Theorem 3, using previous results in Theorem 10.

## G.12 PROOF OF THEOREM 1

**Preliminaries.** Our theorem will use information theory, and in particular will require defining probability divergences. WLOG, we will consider a finite domain $\mathcal{X}$. For a distribution $P$ on on $\mathcal{X}$, and we will abuse notation and also write $P(x) = \Pr_{X \sim P}[X = x]$. For two distributions $P, Q$ on $\mathcal{X}$, the total variation distance is given by

$$TVD(P, Q) = \frac{1}{2} \sum_{x \in \mathcal{X}} |P(x) - Q(x)|.$$

Similarly, the KL-divergence is given by

$$D_{KL}(P\|Q) = \sum_{x \in \mathcal{X}} P(x) \ln(\frac{P(x)}{Q(x)}).$$

The $KL$ divergence satisfies several important properties. The data processing inequality states that, for any randomized process $f$, it holds that $D_{KL}(P\|Q) \geq D_{KL}(f(P)\|f(Q))$ (the TVD also satisfies this inequality). For distributions $P_1, P_2$ on $\mathcal{X}$ and $Q_1, Q_2$ on $\mathcal{Y}$, we may define the conditional KL divergence by

$$D_{KL}(Q_1|P_1\|Q_2|P_2) = \mathbb{E}_{x \sim P_1} \left( \sum_{y \in \mathcal{Y}} Q_1(y|P_1 = x) \ln(\frac{Q_1(y|P_1 = x)}{Q_2(y|P_2 = x)}) \right)$$

The chain rule of KL divergences states that $D_{KL}(P_1, Q_1\|P_2, Q_2) = D_{KL}(P_1\|Q_1) + D_{KL}(Q_1|P_1\|Q_2|P_2)$. It is easy to use this rule that if $P_1, Q_1$ are independent along with $P_2, Q_2$, then $D_{KL}(P_1, Q_1\|P_2, Q_2) = D_{KL}(P_1\|Q_1) + D_{KL}(Q_1\|Q_2)$. For proofs of the above and more information, refer to the information theory textbook (such as doi (2005)).

**Proof.** Suppose to the contrary that there was such a protocol, given by local randomizers $\mathcal{R}_i$. Consider an "honest" world where $G$ has no edges except to user $n$. For $i = 1$ to $n - 1$, let $z_i$ indicate whether the edge from $i$ to $n$ is present, and let $z_i \sim Bern(\frac{1}{2} + p)$ i.i.d. where $p = \frac{m}{4n} + \frac{1}{6\sqrt{n}(e^\epsilon - 1)}$. Let $y_i$ indicate whether $U_i$ acts maliciously, and let $y_i \sim Bern(\frac{m}{2n})$, i.i.d. Finally, suppose that each malicious with an edge to user $n$ behaves as if the edge doesn't exist. Formally, for $i = 1$ to $n - 1$, the random variable

$$x_i^0 = \begin{cases} 0 & y_i = 1 \\ z_i & \text{otherwise} \end{cases}$$

defines the input of user $i$ to the randomizer that concerns edges to $U_n$. Next, $U_n$, who will behave honestly in this world, will use $w_i^0 = z_i$ for $i = 1, \ldots, n-1$, as their input. Thus, the protocol responses in this world will be

$$\left(\mathcal{R}_1(\langle \mathbf{0}, x_1^0 \rangle), \ldots, \mathcal{R}_{n-1}(\langle \mathbf{0}, x_{n-1}^0 \rangle), \mathcal{R}_n(\langle w_1^0, \ldots, w_{n-1}^0 \rangle)\right),$$

where $\mathbf{0}$ indicates a vector of $n-1$ 0s. Let $r_{(n-1)}^0$ indicate the first $n-1$ terms of the above tuple, and $r_n^0$ indicate the final term. By the honest error guarantee, the responses may be post-processed into an estimate $\hat{d}_n(r_{(n-1)}^0, r_n^0)$ such that $|\hat{d}_n - d_n| \leq \frac{m}{4} + \frac{\sqrt{n}}{40\epsilon}$ for each graph $G$. With probability at least 0.9, $U_n$ has degree at least $\frac{n}{2} + \frac{m}{4} + \frac{\sqrt{n}}{6\epsilon} - 3\sqrt{n}$ (since $\epsilon < \frac{1}{20}$) in this world. Thus, $\Pr[\hat{d}_n \geq \frac{n}{2}] \geq 0.8$, where the probability considers the randomness in $G$, and running the protocol described above.

In the second, "malicious" world, $G$ will be defined the same way, except each $z_i \sim Bern(\frac{1}{2} - p)$ i.i.d. For $i = 1$ to $n-1$, malicious users will be chosen the same way as before, and we may define $y_i$ as before. This time, each malicious user will behave as if there is an edge to user $n$, so they will use

$$x_i^1 = \begin{cases} 1 & y_i = 1 \\ z_i & \text{otherwise} \end{cases}$$

as their input to $U_n$. $U_n$ will act maliciously as well, by acting as if he has an edge to each malicious user, and with probability $s = 1 - \frac{(1-2p)}{(1+2p)(1-m/2n)}$, to an honest user as well. Formally, he will define $y_i' \sim Bern(s)$, and compute

$$w_i^1 = \begin{cases} 1 & y_i = 1 \text{ or } y_i' = 1 \\ z_i & \text{otherwise} \end{cases}$$

Thus, the output of the protocol in this world will be

$$\left(\mathcal{R}_1(\langle \mathbf{0}, x_1^1 \rangle), \ldots, \mathcal{R}_{n-1}(\langle \mathbf{0}, x_{n-1}^1 \rangle), \mathcal{R}_n(\langle w_1^1, \ldots, w_{n-1}^1 \rangle)\right).$$

Let $r_{(n-1)}^1$ indicate the first $n-1$ terms of the above tuple, and $r_n^1$ indicate the final term. From the maliciour error guarantee, we have that in this world, $\Pr[\hat{d}_n = \perp \vee \hat{d}_n \leq \frac{n}{2}] \geq 0.8$. This is a disjoint event from the event in the honest world, and in particular, it implies that $TVD((r_{(n-1)}^0, r_n^0) \| (r_{(n-1)}^1, r_n^1)) \geq 0.8$.

However, observe that each $w_i^0$ is identically distributed to each $w_i^1$—they are both drawn from $Bern(\frac{m}{2n} + (1 - \frac{m}{2n})(\frac{1}{2} + p))$. In the honest world, we have that $\Pr[x_i^0 = 0 | w_i^0 = 0] = 1$, and $\Pr[x_i^0 = 0 | w_i^0 = 1] = \Pr[y_i = 1]$ since the only way $x_i^0 = 0$ can occur if $z_i^0 = 1$ is if $y_i = 1$.

Similarly, in the malicious world we have $\Pr[x_i^1 = 0 | w_i^1 = 0] = 1$, and

$$\Pr[x_i^1 = 0 | w_i^1 = 1]$$
$$= \frac{\Pr[x_i^1 = 0, w_i^1 = 1]}{\Pr[w_i^1 = 1]}$$
$$= \frac{\Pr[y_i = 0, y_i' = 1, z_i = 0]}{1 - \Pr[y_i = 0, y_i' = 0, z_i = 0]}$$
$$= \frac{2p - \frac{m}{4n} + p\frac{m}{2n}}{1 - (1 - m/2n)(1 - s)(1/2 + p)}$$
$$= \frac{2p - \frac{m}{4n} + p\frac{m}{2n}}{1/2 + p}$$

Now, we will derive a contradiction using the information between the two worlds. In either world, we have that $r_n$ is a post-processing of $w_{(n-1)}$. By the data processing inequality and chain rule of KL divergences, we have

$$D_{KL}((r_{(n-1)}^0, r_n^0) \| (r_{(n-1)}^1, r_n^1))$$
$$\leq D_{KL}((r_{(n-1)}^0, w_{(n-1)}^0) \| (r_{(n-1)}^1, w_{(n-1)}^1))$$
$$= D_{KL}(w_{(n-1)}^0 \| w_{(n-1)}^1) + D_{KL}(r_{(n-1)}^0 | w_{(n-1)}^0 \| r_{(n-1)}^1 | w_{(n-1)}^1).$$

The first term is clearly 0. Using conditional independence of the $x_i$s and the protocols, we can write

$$D_{KL}(r^0_{(n-1)}|w^0_{(n-1)}\|r^1_{(n-1)}|w^1_{(n-1)})$$

$$= \sum_{i=1}^{n-1} D_{KL}(r^0_i|w^0_i\|r^1_i|w^1_i).$$

Now, we apply Theorem 1 of Duchi et al. (2013), which states that, when $\mathcal{R}_i$ satisfies local differential privacy, and $x^0_i, x^1_i$ are two distributions, we have $D_{KL}(\mathcal{R}_i(x^0_i)\|\mathcal{R}_i(x^1_i)) \leq 4(e^\epsilon - 1)^2 TVD(x^0_i, x^1_i)^2$. Plugging in the above probabilities, we have

$$TVD(x^0_i|w^0_i = 0, x^1_i|w^1_i = 0) = 0$$

$$TVD(x^0_i|w^0_i = 1, x^1_i|w^1_i = 1) = \left| \frac{2p - \frac{m}{4n} + p\frac{m}{2n}}{1/2 + p} - \frac{m}{2n} \right|$$

$$= \frac{|2p - m/2n|}{1/2 + p}.$$

Plugging in $p = \frac{m}{4n} + \frac{1}{6\sqrt{n}(e^\epsilon - 1)}$, both of the above are at most $\frac{1}{3\sqrt{n}(e^\epsilon - 1)}$, and thus $D_{KL}(r^0_i|w^0_i\|r^1_i|w^1_i) \leq \frac{4}{9n}$. Thus, we have

$$D_{KL}(r^0_{(n-1)}|w^0_{(n-1)}\|r^1_{(n-1)}|w^1_{(n-1)}) \leq \frac{4}{9}.$$

Finally, by Pinsker's inequality, we know that

$$TVD((r^0_{(n-1)}, r^0_n), (r^1_{(n-1)}, r^1_n)) \leq \sqrt{\frac{2}{9}} < 0.5,$$

completing the contradiction.