# OpenReview forum: "Robust Locally Differentially Private Graph Analysis"
_ICLR.cc/2025/Conference — Submitted to ICLR 2025_

### Official Review · Reviewer_mTXY · 2024-10-31

**Soundness:** 3
**Presentation:** 2
**Contribution:** 2
**Rating:** 8
**Confidence:** 3

**Summary:**

This paper continues a line of study on exploring the impact of poisoning attack in local differential privacy. In particular, they consider the task of estimating the degrees of each vertex under the widely used notion of edge-level DP, in which two graphs are considered neighboring if they differ in one edge. For the poisoning setting, they consider two types of attack. First is the input poisoning, where a malicious user falsify their underlying input. A stronger one is the response poisoning, where the adversary has access to the implementation of the LDP randomizer.

Under such settings, they first show that the navie implementation of the Laplace mechanism or the Randomized Response mechanism leads to almost trivial gurantee on the soundness. Then, by revealing the fact that the information are naturally redundant for degree estimation, they design a verification mechanism to improve the soundness under poisoning attack, and achieving $O(m(1+1/\varepsilon) + \sqrt{n}/\varepsilon)$ accuracy and soundness with a small failure probability, based on the randomized response mechanism. Finally, they combining the laplace mechanism and improve the the accuracy to logarithmic error for "honest" users.

**Strengths:**

1. The technical lemmas and theorems in this paper are clearly stated and correct.
2. The hybrid mechanism for reducing the error is interesting.

**Weaknesses:**

I agree that it is natural to consider the poisoning attack within the context of local DP, and the edge-level (global) differential privacy is a rather standard notion. However, I think using edge-DP in the local DP model is unusual. In particular, I agree that "the users do not explicitly share this information; rather, it is implicitly shared by the structure of the graph itself." My concern, however, is whether studying local differential privacy remains meaningful, given that the graph's structure may *already* "leak" information to other users within it.

In the last review process, I mentioned a typo in Appendix G.3 (in line 1325 of this version) that it should be $|L_i|\leq \frac{1}{\varepsilon}\ln \frac{n}{\delta}$ instead of $|L_i|\leq \frac{1}{\varepsilon}\ln \frac{\delta}{n}$. But the typo seems to be still exist in this version, so I worry that the authors did not tidy up their proofs carefully.

**Questions:**

The authors have answered my questions in the last review process.

---

> ### Author Response · Authors · 2024-11-21
> **Rebuttal**
>
> We thank the reviewer for their insightful comments and address their main concerns below. We are glad to learn that we were able to address the reviewer's concerns with our previous rebuttal. Here, we would like to take the opportunity to provide some additional context as to why edge-LDP is the most meaningful privacy guarantee in our setting.
>
> There are two standard notions of DP for graphs - edge DP where the information about a single edge is protected, and node-DP where the information about an entire node of the graph is protected. Although node-DP is a stronger privacy guarantee, it is not suitable for our setting. Recall that our goal is to estimate the degree of every user. Now protecting this under node-LDP would require adding noise proportional to $\frac{n}{\epsilon}$ (since under node-LDP all the edges of a user can change resulting in a sensitivity of $n$). But this means that the amount of noise itself is much higher than the true answer (a  degree can be at most $n$ and for real-world graphs it is often much lower) rendering the estimates to be completely meaningless.
> As a result, the current literature considers edge-DP to be the standard notion of privacy in the local setting and has been used in the context of a variety of tasks such as counting the number of triangles [a], k-core decomposition [b], training graphical neural networks [c] and synthetic graph generation [d].
>
>
> [a] Triangle counting with local edge differential privacy
>
> [b]  Near-Optimal Differentially Private k-Core Decomposition
>
> [c] LPGNet: Link Private Graph Networks for Node Classification
>
> [d]  Generating Synthetic Decentralized Social Graphs with Local Differential Privacy.

---

> > ### Comment · Reviewer_mTXY · 2024-11-22
> >
> > Thanks very much for your response! Your rebuttal is basically saying that for the problem you study, node-level DP does not give non-trivial utility guarantee, while I am already aware of this. I raised the concern about the setting because it feels counterintuitive to me to study local differential privacy in a context where some local data is already shared with others due to the graph structure. But after reading the settings in some of the references you mentioned in your response, I am sufficiently convinced that this should not be an issue —— sharing data with users in the network does not equate to placing trust in a central aggregator, so local DP is still meaningful. So I take this concern back.
> >
> > After convincing myself that there are no critical issues with the setting and motivation, I carefully went through the entire paper once again. To be honest, I am not particularly impressed by this paper. From a technical perspective, the algorithms are built upon basic mechanisms, and much of the analysis heavily relies on standard concentration inequalities. However, from a perspective of appreciation, of course I agree that all these contributions are still highly non-trivial, and it is the first work studying the poisoning attack for graphs under LDP. I also found that the $O(m+\sqrt{n}/\varepsilon)$ critical point of the honest error and malicious error described in Theorem 1 is pretty interesting.
> >
> > Overall speaking, I would not fight for accepting this paper but I believe its conceptual contribution should merit the acceptance to ML conferences like ICLR. Therefore, I decided to raise my score to 8.

---

> > > ### Author Response · Authors · 2024-11-26
> > > **Thank You**
> > >
> > > We would like to sincerely thank the reviewer for engaging with us in good faith and finding our contributions to be "highly non-trivial".

---

### Official Review · Reviewer_Sabb · 2024-11-03

**Soundness:** 3
**Presentation:** 2
**Contribution:** 3
**Rating:** 6
**Confidence:** 3

**Summary:**

This work introduces a systematic framework for analyzing poisoning attacks in Local Differential Privacy (LDP) protocols for graph degree estimation. The authors propose two key metrics: honest error and malicious error, to quantify the impact of adversarial manipulation on both honest users and overall estimation accuracy. Their analysis reveals that poisoning attacks are more effective when targeting randomized response mechanisms compared to direct input manipulation. The work contributes two novel attack vectors: degree inflation and degree deflation, providing a comprehensive examination of potential adversarial strategies. To counter these threats, the authors leverage the inherent redundancy in graph structures—specifically, the property that edges are naturally reported by both connected vertices—to develop two defensive protocols. The empirical evaluation encompasses both synthetic and real-world (Facebook) datasets of varying scales, demonstrating the effectiveness of their findings and proposed defenses. Their results provide important insights into the vulnerability of LDP protocols in graph statistics and offer practical approaches for enhancing robustness against poisoning attacks.

**Strengths:**

**Originality:**
The paper presents the first comprehensive study exploring poisoning attacks in LDP protocols for graph degree estimation. The work introduces several novel ideas. These include:
-Demonstrating that poisoning attacks on randomized responses (i.e. output of the noise addition required for LDP) are more effective than input data poisoning
- Leveraging edge-sharing properties between adjacent nodes for malicious user detection
- Developing a method to distinguish between LDP-induced and malicious inconsistencies
- Proposing solutions that exploit the inherent redundancy in graph edge reporting for attack mitigation

**Quality:**
The paper demonstrates technical soundness through:
- An appropriate mathematical formulation of the real-world graph degree estimation problem
- Rigorous analysis of dual sources of edge distribution inconsistency: LDP randomization and malicious manipulation
- Comprehensive parameter evaluation across privacy budget ($\epsilon$), accuracy error, malicious error, database size, and adversary size and bounds

**Clarity:**
The work presents its ideas through:
- Practical motivation grounded in real-world applications, particularly social network influence analysis (e.g., Mastodon)
- Systematic development of robust degree estimation protocols that address both malicious and honest errors

**Significance:**
The paper makes several significant contributions:
- Direct applicability to real-world scenarios of influencer detection and manipulation in social networks
- A good (but perhaps not comprehensive in terms of data sources) empirical validation using both synthetic and Facebook datasets
- Practical defensive measures for preventing adversaries from promoting malicious users as influential nodes

The work provides both theoretical insights and practical defensive measures against poisoning attacks in LDP protocols for graph analysis. The comprehensive parameter analysis and thorough experimental validation across multiple datasets demonstrate both the theoretical and practical significance of the contributions.

**Weaknesses:**

**Writing and Technical Issues:**
- There is redundant wording in line 122, page 3: "distributed graphs and has been widely studied widely"
- The reference formatting lacks consistency throughout the paper. For instance:
1. Author names are inconsistently abbreviated (e.g., "Xiaoyu Cao, et al." vs. full author lists)
2. Conference/journal names and their formatting vary (e.g., inconsistent capitalization and abbreviations)
3. In the current version, the latest reference is from the year 2022; The reference section could be strengthened by including recent (2023-2024) developments in LDP poisoning attacks, particularly works on LDP protocol robustness and defense mechanisms against output poisoning. This additional context would further highlight the paper's pioneering contribution to LDP-protected graph poisoning attacks. A list that is far from exhaustive is given below. Other references have been updated but not reflected as such: e.g., Li et al. (2022) on fine-grained poisoning attacks has appeared in a more final form at USENIX Security 2023.

**Figures and Visualizations:**
1. Figure Quality:
- Figures 3 and 4 are not provided in vector format, resulting in poor scalability and reduced readability when zoomed
- The font styles and sizes in subcaptions (a)(b)(c)(d) lack consistency across Figures 3 and 4, etc..
2. Experimental Design and Presentation:
- A limitation in the experimental design appears in Figure 4, where the varying database sizes (m=1332 vs m=1320) lack rigorous theoretical motivation. The authors' justification that these parameters "meet the asymptotic theoretical error bounds" requires more substantial analytical support to establish the connection between these specific numerical choices and the theoretical foundations.
- The choice of $\epsilon$ values (0.7 and 3.00) requires justification
- Consider using other additional visualization methods for the comparative analysis, as it might better highlight the differences in some malicious errors and honest errors.

**References:**
1. Huang, Kai, Gaoya Ouyang, Qingqing Ye, Haibo Hu, Bolong Zheng, Xi Zhao, Ruiyuan Zhang, and Xiaofang Zhou. "LDPGuard: Defenses against data poisoning attacks to local differential privacy protocols." IEEE Transactions on Knowledge and Data Engineering (2024).
2. Sun, Xinyue, Qingqing Ye, Haibo Hu, Jiawei Duan, Tianyu Wo, Jie Xu, and Renyu Yang. "Ldprecover: Recovering frequencies from poisoning attacks against local differential privacy." arXiv preprint arXiv:2403.09351 (2024).

**Questions:**

**Venue Fit and Positioning:**
While the paper presents solid technical contributions in security and privacy, its fit with ICLR's focus on learning is not immediately clear. Privacy/security of ML is certainly on topic for ICLR, however it would be appreciated if the authors elaborate on their thoughts here, and whether they had considered a security/privacy venue. Given that  many cited works on LDP and poisoning attacks appear in security and privacy venues.

**Technical Clarifications:**
- The finding that "the rate of flagging is less aggressive for FB since it is a sparse graph" (line 508) is not readily apparent in Figure 3. Could the authors clarify this observation with supporting evidence?
- How does the computational complexity of the proposed protocols scale with very large graphs? Are there any limitations or performance bottlenecks?
- For the experiments comparing input poisoning and response poisoning, what informed the choice of different database sizes (m=1332 vs m=1320)? How do these specific values relate to the theoretical bounds?
- The paper uses an argument about the Bernoulli distribution to distinguish between LDP-induced and malicious inconsistencies. It would be appreciated if the authors might elaborate on the theoretical justification here; The sensitivity of this modeling choice to different graph topologies (beyond the tested Facebook and synthetic datasets) and different attack patterns. How might the results be affected by: networks with heterogeneous degree distributions, social networks exhibiting power-law connectivity, and graphs with varying density across different regions?

---

> ### Author Response · Authors · 2024-11-21
> **Rebuttal**
>
> We thank the reviewer for their insightful comments and address their main concerns below. We are very glad to know that the reviewer found our work to provide both theoretical insights and practical implementations. We would like to take the opportunity to address the main concerns below.
>
> **Fit.** We chose ICLR  since it has long been a leading venue for research in differential privacy. Here are a few examples of papers from ICLR 2024
>
> a. A Differentially Private Clustering Algorithm for Well-Clustered Graphs
>
> b. Numerical Accounting in the Shuffle Model of Differential Privacy
>
> c. Privacy Amplification for Matrix Mechanisms
>
> d. Efficiently Computing Similarities to Private Datasets
>
> **Rate of flagging.** The quoted statement was not made in reference to Fig. 3, but rather in the context of the immediately preceding discussion (lines 504-507), which we reproduce here for clarity.
>
> Our protocols are able to flag malicious
> users when they target a large number of honest users. Specifically, for the strongest degree deflation
> attack, Hybrid flags 4.5% and 49.8% of the malicious users for FB and Syn, respectively. RRCheck,
> on the other hand, flags 3% and 59.3% of the malicious users for FB and Syn, respectively.  Note that
> the number of actual honest users affected by a malicious user is bounded by its degree. This is the reason why
> rate of flagging is less aggressive for FB since it is a sparse graph (the maximum degree is low).
> The rates of flagging are  presented in Table 1.
>
> **Choice of parameters for experiments.**  All our theoretical results are completely *general purpose* meaning they do not rely on any assumptions about problem-specific parameters, such as the privacy parameter, the underlying graph, input distribution, or the type of attack. This means that our theoretical results hold for *any arbitrary* choice of these parameters.
>
>   For the experiments in the main body we considered two settings for the privacy parameter (i) high privacy regime with $\epsilon=0.7$ -  (values $\epsilon < 1$ is  considered to provide high privacy guarantees), and (ii) $\epsilon=3$ for low privacy. Similarly, we considered two settings for the number of malicious users, $m$, $m=$1% (low poisoning rate) and $m=$33% (high poisoning rate). $m=$33% for our two datasets FB and Syn gives the concrete numbers  m=1332 and m=1320, respectively. Prior work has shown that $m=$1% is considered a realistic threat in practice [28]. $m=$33% corresponds to $\frac{1}{3}$  of parties being malicious which is a classic threshold considered in the literature on Byzantine robustness  and cryptography [a][b][c].  Additional experiments with different choices of parameters are presented in Appendix D.
>
> [28] Back to the drawing board: A critical evaluation of poisoning attacks on production federated learning
>
> [a] Asynchronous consensus and broadcast protocols, 1985
>
> [b] How to Play Any Mental Game, 1987
>
> [c] Asynchronous Secure Computations, 1999
>
> **Impact of different problem specific parameters** - One of the biggest advantages of our robustness guarantees is that they are completely *general purpose* and *attack agnostic* -- i.e., our guarantees are completely unaffected by the choice of problem-specific parameters, such as the privacy parameter, the underlying graph, input distribution, or the type of attack. Our consistency check is based on an analysis of the tail bound for the Bernoulli distribution, which stems from the properties of the classic LDP mechanism, Randomized Response -- the individual bits of the user's adjacency lists are Bernoulli random variables. This analysis is again independent of everything else about the problem setup.
>
> **Computational complexity.** Our algorithm requires O(n) work to estimate the and the operations are extremely light weight.
>
> **Related Work.** We will update the related work section with more concurrent work. However, we would like to highlight that all prior work still focuses on tabular data setting, the novelty of our setting is that this is the *first* work studying the impact of poisoning for graphs under LDP.

---

> > ### Comment · Reviewer_Sabb · 2024-11-25
> > **Response to rebuttal**
> >
> > Thanks to the authors for their rebuttal.
> >
> > > **Fit** We chose ICLR since it has long been a leading venue for research in differential privacy. Here are a few examples of papers from ICLR 2024 ...
> >
> > I acknowledge the authors have found ICLR papers that focus on privacy. This is helpful. I acknowledge that the paper is not out of scope for ICLR, but perhaps might have slightly narrow appeal to the ICLR community.
> >
> > > **Rate of flagging.** The quoted statement was not made in reference to Fig. 3, but rather in the context of the immediately preceding discussion (lines 504-507), which we reproduce here for clarity...
> >
> > The authors are essentially repeating what's already in lines 504-507. While they mention Table 1 in this paragraph, the table itself is placed in the appendix. The paper would be well served by referencing Table 1.
> >
> > > **Choice of parameters for experiments.** All our theoretical results are completely general purpose meaning they do not rely on any assumptions about problem-specific parameters, such as the privacy parameter, the underlying graph, input distribution, or the type of attack.
> >
> > I understand the authors chose $\epsilon = 0.7$ as a high privacy setting, but am curious about the specific reason for this value. From the range of possible values $0.1, 0.2, ..., 3$, why exactly $0.7$? Was this choice made to make the visualizations in figures (e.g., Fig. 7) more distinguishable and clear? Or was it randomly selected just to represent a high privacy level? Did the authors consider how other values like $0.1$ or $0.3$ might affect the visualization of their results?
> >
> > There seems to be an inconsistency in the numbers: The authors state that for the Facebook dataset with 4082 users, using a 33% poisoning rate should result in 1347 malicious users. However, the paper reports 1332 malicious users. Could the authors please clarify this discrepancy?
> >
> > > **Impact of different problem specific parameters** - One of the biggest advantages of our robustness guarantees is that they are completely general purpose and attack agnostic ...
> >
> > No further issues here
> >
> > > **Computational complexity.** Our algorithm requires O(n) work to estimate the and the operations are extremely light weight
> >
> > The authors demonstrate their protocol on Facebook and synthetic graph datasets. It would be helpful to understand how it performs in terms of actual running time? Specifically:
> > What is the computational time for normal operation (without attacks)?
> > How much additional time is required when handling poisoning attacks?
> > Using the Apple phone call example, how long would it take for poisoned users to affect the graph collection process compared to the regular case?
> >
> > > **Related Work.** We will update the related work section with more concurrent work. However ...
> >
> > The review was not asking the authors to cite specific papers - it was just noticed that the literature review stops at 2022. An updated review through 2024 would help verify novelty claims about being the first graph-based approach and might reveal useful insights from recent research that could strengthen this paper, especially since the graph-based approach builds upon some ideas from tabular data work.
> >
> > The typos and figure issues that were identified in the previous review have not been fixed in this version.
> >
> > While the rebuttal, updates, and discussion with reviewers as a whole are appreciated, I will maintain my scores/overall ratings.

---

> > > ### Author Response · Authors · 2024-11-26
> > > **Rebuttal**
> > >
> > > We thank the reviewer for engaging with us and provide additional details:
> > >
> > > ** Figures.** We will fix all the typographical issues and we will include Table 1 in the main body.
> > >
> > > **Related work.** We thank the reviewer for the suggestion and would include an updated review through 2024.
> > >
> > > **Choice of experimental parameters.** We selected $\epsilon=0.7$ at random in the high privacy regime. The visualization remains the same for any value in this regime. There is a typo -- the number of nodes in FB should be 4039 (as noted in the original paper [a]). We thank the reviewer for catching this and will fix this typo in the paper.
> > >
> > > [a] Learning to discover social circles in ego networks
> > >
> > > **Computational complexity.** The computational cost is roughly twice that of the naive approach based on randomized response. We will revise our paper to also include the runtime numbers.
> > > The clients do not require any extra time to perform poisoning attacks. Since our protocols are non-interactive, clients can deceive the server in a single, one-shot communication.

---

### Official Review · Reviewer_rYuk · 2024-11-03

**Soundness:** 2
**Presentation:** 1
**Contribution:** 2
**Rating:** 3
**Confidence:** 3

**Summary:**

The paper explores the vulnerability of locally differentially private (LDP) graph analysis to poisoning attacks, where adversaries skew results by submitting malformed data. The authors highlight that LDP protocols are particularly susceptible to such attacks and leverage the natural redundancy in graph data to design robust degree estimation protocols under LDP. They propose a formal framework to analyze protocol robustness, focusing on accuracy for honest users and soundness for malicious ones. The paper introduces new protocols that significantly reduce the impact of adversarial poisoning and computes degree estimates with high utility. Comprehensive empirical evaluations on real-world datasets validate the effectiveness of these protocols. The study contributes to the understanding of poisoning attacks under LDP and provides practical solutions for more secure graph analysis.

**Strengths:**

The paper focuses on an interesting research question and builds on strong theoretical foundations, including information theory and differential privacy, to establish lower bounds and prove the efficacy of the proposed solutions.

**Weaknesses:**

My fundamental concern lies in that the practical significance of the paper is rather unclear. The paper gives an motivating real-world example, which involves degree collection on social networks. In practice, social networks often publicly display the number of followers or connections a user has, rendering the need for private degree aggregation obsolete.

**Questions:**

If the major focus of the paper more targeted to aggregated degree calculation or network publishing?

---

> ### Author Response · Authors · 2024-11-18
> **Rebuttal**
>
> We thank the reviewer for their insightful comments and address the main concerns below.
>
> **Motivation** We work in the setting of a distributed graphs -- i.e., nobody has access to the entire graph. Hence, if the graph is in the context of a distributed social media network, such as Mastodon, bluesky, it is *impossible* for the server to publish the exact degrees of the users because it simply does not have access to this information. The only way the server can get access to this information is collecting this directly from the users. Now, one of the biggest selling point of distributed social networks is privacy, hence, it is unrealistic to imagine that the users will be okay to report their degrees in the clear to the server which showcases the real-world applicability of our protocol.
> Another example of a distributed graph is in the context of phone call graphs. Consider that every iPhone owner is a user or node, and an edge between two users indicates a phone call between them. Apple, acting as the untrusted aggregator, wants to compute a degree vector of the entire graph. The edges are sensitive (phone calls reveal users' personal social interactions), so users cannot submit their data to Apple directly. Instead, they add noise to their data to achieve a local differential privacy guarantee before sharing it with Apple.
>
> Studying privacy-preserving degree distribution is a fundamental and classic problem in the graph privacy literature and has been examined thoroughly in prior literature starting from the seminal works of Nissim [a], Hay et al. [b], Karwa et al. [c]. We extend this body of work to a new dimension  by considering the threat of poisoning attacks which is again extremely realistic in our setting.
>
>
> [a] Smooth Sensitivity and Sampling in Private Data Analysis, Kobbi Nissim, Sofya Raskhodnikova, Adam Smith
>
> [b] Accurate Estimation of the Degree Distribution of Private Networks, Michael Hay, Chao Li, Gerome Miklau, David Jensen
>
> [c] Private Analysis of Graph Structure , Vishesh Karwa, Sofya Raskhodnikova, Adam Smith, Grigory Yaroslavtsev

---

> > ### Author Response · Authors · 2024-11-26
> > **Checking In**
> >
> > Dear Reviewer,
> >
> > We wanted to check in if there are additional concerns that we can help address.
> >
> > Thanks, Authors

---

> > > ### Comment · Reviewer_rYuk · 2024-12-02
> > >
> > > I appreciate the authors for addressing my concern about the motivation of the work. I would like to suggest the authors to add the motivating examples to the revised paper to make its motivation and potential contribution more clear.
> > > I would also ask more clarification on how would the authors differentiate their work with the private network publishing works?

---

> > > > ### Author Response · Authors · 2024-12-02
> > > > **Rebuttal**
> > > >
> > > > We are glad to be able to address the reviewer's concern and will include this discussion in the paper.
> > > >
> > > > **Difference from prior work** - Ours is the *first work* to study the impact of poisoning attacks for graphs under LDP and furthermore, to provide the *first* provably robust algorithms for graph statistics.  All prior work on private graph analysis has focused on computing different graph statistics *only* under privacy -- none of them consider poisoning attacks. Additionally, all prior studies on data poisoning under LDP have been limited to tabular or key-value data.

---

### Official Review · Reviewer_BeqU · 2024-11-03

**Soundness:** 2
**Presentation:** 2
**Contribution:** 2
**Rating:** 3
**Confidence:** 4

**Summary:**

This paper studies the problem of data poisoning attacks to graph data analysis under local differential privacy, specifically targeting the estimation of node degree distribution. Although the studied problem is important, the contribution is incremental, and the proposed solution, along with its theoretical analysis, contains flaws.

**Strengths:**

1. The studied problem is important.

2. Extensive theoretical analysis is provided.

**Weaknesses:**

1. The graph data perturbation involved in this work does not satisfy LDP. This work is based on edge LDP, which protects the existence of an edge between any two users. In terms of adjacency vector, the sensitivity of an edge’s existence should be 2 bits. Thus, when applying RR to perturb that vector, the probability should be $\frac{1}{1+e^{\epsilon/2}}$, rather than $\frac{1}{1+e^\epsilon}$. In terms of degree perturbation, the sensitivity of an edge’s existence should be 2, as the edge connects to two nodes and affects the degree of both nodes. Thus, when applying Laplace noise, it should be $Lap(2/\epsilon)$, rather than $Lap(1/\epsilon)$. This issue has been widely studied in the literature [1-2].

[1] Liu Y, Wang T, Liu Y, et al. Edge-Protected Triangle Count Estimation under Relationship Local Differential Privacy. IEEE Transactions on Knowledge and Data Engineering, 2024.

[2] Ye Q, Hu H, Au M H, et al. LF-GDPR: A framework for estimating graph metrics with local differential privacy. IEEE Transactions on Knowledge and Data Engineering, 34(10): 4905-4920, 2022.

2. The contribution is incremental. The difference between input poisoning and output poisoning in the context of LDP has been thoroughly studied in the literature. In addition, it is unclear how the honest error differs from the malicious error. Can the authors provide a concrete example for illustration?

3. The experimental evaluation needs to be improved. It is unclear what observation and conclusion can be made from Figure 4.

4. The presentation needs to be improved. There are quite a few typos in the manuscript. Here are some examples.
- In page 2, “upto” -> “up to”
- In page 4, “reponse” -> “response”
- In page 5, “In our first scenario, consider” -> “Our first scenario considers”

**Questions:**

Please refer to the Weaknesses section.

---

> ### Author Response · Authors · 2024-11-18
> **Rebuttal**
>
> We thank the reviewer for their insightful comments and address the main concerns below.
>
> **Factual Error.** Our analysis is indeed correct and in line with the definition of edge-LDP which was introduced in the seminal paper by Nissim et al. [a] . What the reviewer points to is not edge-LDP but **relationship**-DP which was introduced by Imola et al in [b][c]. Nevertheless, translating between these two definitions is straightforward - any $\epsilon$-edge LDP protocol satisfies $2\epsilon$-relationship DP (Proposition 1 in  [b]). Note since the privacy definitions only affect a constant term all our asymptotic conclusions from our theoretical results are completely unaffected by the choice of the privacy definition.
>
> [a] Smooth Sensitivity and Sampling in Private Data Analysis, Kobbi Nissim, Sofya Raskhodnikova, Adam Smith
>
> [b] Locally differentially private analysis of graph statistics 2021, Jacob Imola, Takao Murakami, Kamalika Chaudhuri
>
> [c] Communication-Efficient triangle counting under local differential privacy 2022, Jacob Imola, Takao Murakami, Kamalika Chaudhuri
>
>
> **Contributions.** Ours is the *first* work to study the impact of poisoning under LDP for graphs -- prior work only focused on tabular data or key-value datasets. Nevertheless, showing the separation between input and response poisoning the is *not* at all our primary contribution. Our major contributions are
> 1. Providing a new framework for quantifying robustness
> 2. A new **lower bound** result on poisoning attacks for graphs (Thm. 1)
> 3. Providing the first provably robust degree estimation protocol that is completely **attack agnostic** and **optimal** (i.e., matches the above lower bound)
>
> All of these theoretical contributions are completely novel and highly non-trivial. In fact, ours is the first work to give *provable* and *attack-agnostic* robustness guarantee against any LDP protocols, graphs or otherwise -- prior defenses (all in the context of tabular data) were empirical and customized to specific attacks.
>
> **Honest and Malicious Error.**  Honest error corresponds to the error introduced in the degree estimate of an *honest client* while malicious error corresponds to the error introduced in the degree estimate of a *malicious client*. In a nutshell, honest error and malicious error quantify the error of the two disjoint sets of clients.
>
> **Experiments.** We have carried out an extremely extensive experimental evaluation with **16** different attacks capturing real-world attack scenarios. Due to lack of space, we could only include a subset of our experimental results in the main paper -- an additional set of results are presented in Appendix D. Fig. 4 plots both malicious and honest error for 1) the naive baseline 2) our proposed protocols. As seen from the plots, the main observation is that the both honest and malicious error is significantly lower for our proposed protocols. Additionally, the plot also shows the separation between input and response poisoning.  As such, the plot empirically demonstrates the superiority of our proposed protocols in protecting against poisoning attacks and thereby validate our theoretical results.
> Could the reviewer please point out how can the evaluation be improved?
>
> **Presentation.** We will do a thorough copy-edit pass of the paper.

---

> > ### Author Response · Authors · 2024-11-26
> > **Checking In**
> >
> > Dear Reviewer,
> >
> > We wanted to check in if there are additional concerns that we can help address.
> >
> > Thanks,
> > Authors

---

### Meta-Review · Area_Chair_ThcD · 2024-12-14

**Metareview:**

# Summary of Contribution

This paper studies a setting where each user is a node in a graph and has its adjacency list. The goal is to compute the estimate degree of each user while respective edge differential privacy (edge DP). The standard protocol for this problem is to add Laplace noise to each degree and publish it, which achieves error of $O(1/\epsilon)$. However, this protocol is very non-robust: A malicious user can lie and make their estimate degree arbitrarily large. The contributions of this paper are as follows:
- They propose a model for robust protocols in edge-LDP setting, together with the concept of honest accuracy and malicious accuracy.
- They provide a protocol that has honest accuracy of roughly $O(1/\epsilon)$ and malicious accuracy of roughly $O(m + \sqrt{n} / \epsilon)$ where $m$ denote the number of malicious users. The protocol is roughly as follows: in addition to the Laplace mechanism, each user also sends their adjacency list, privatized via Randomized Response. We then calculate the estimated degrees in two ways: (i) via the published degree (from Laplace mechanism) by that node, and (ii) via the randomized adjacency lists of the other nodes. If the two are close enough, the estimate is set to (i). Otherwise, it is set to (ii).
- They prove a matching lower bound on the errors.

# Strengths

- **Novel model**: This is the first paper that studies robustness in edge-DP model.

- **Elegant Protocols**: The protocols are based on simple ideas and are well explained in the paper.

- **Matching Lower Bounds**: The authors also show that their lower bounds are nearly optimal by providing lower bounds.

# Weaknesses

- **Importance**: The problem studied / techniques proposed in this paper are quite specific. Namely, it only works in the setting where each piece of data (i.e. edge) is redundant. This is very specific to degree estimation in the edge-LDP model. The methods proposed in this paper are also relatively straightforward from the technical standpoint. Thus, it is unclear how the insights in this paper can lead to broader insights.

- **Practicality**: There are several factors that question the practicality of the protocol:
  - **Error**: The error for a malicious user here is at least $\sqrt{n}/\epsilon$. This means that, even if there is a single malicious user with degree zero, they can pretend to have a degree of $\sqrt{n}/\epsilon$. This is already quite a large. (E.g. on Twitter, this would easily put them in the 0.01% top users.) Of course, as shown by the lower bound, this is inevitable; but this also suggests that maybe edge-LDP is *not* the right model when it comes to robustness.
  - **Communication**: Since every user needs to apply randomized response over the entire $n$-bit vector, the total communication here is (at least) $\Omega(n^2)$. It is thus unlikely that this can be applied to any real-world social network graphs of today.
  - **Parameter setting**: The setting of the threshold $\tau$ involves knowing (or approximating) the number of malicious users $m$ beforehand.

# Recommendation

Although this paper makes a solid theoretical contribution towards the degree estimation problem with edge-LDP, the model might be too specific and not sufficiently practical for a broader audience at ICLR. As such, the paper might be more suitable to be published at a more focused venue (e.g. on privacy / security or distributed graph analysis). Given this, we recommend rejection.

**Additional Comments On Reviewer Discussion:**

The authors give some examples where edge-LDP setting makes sense and also discuss differences compared to previous work. However, the concerns in meta-review remained. Additionally, the authors tried to clarify a misunderstanding with reviewer BeqU, who unfortunately didn't reply during the rebuttal period; nonetheless, I had already taken this into account when recommending rejection.

---

### Decision · Program_Chairs · 2025-01-22

Reject